# Effects of two different biogenic emission models on modelled ozone and aerosol concentrations in Europe

Jianhui Jiang[1], Sebnem Aksoyoglu[1], Giancarlo Ciarelli[2,*], Emmanouil Oikonomakis[1], Imad El-Haddad[1], Francesco Canonaco[1], Colin O'Dowd[3,4], Jurgita Ovadnevaite[3,4], María Cruz Minguillón[5], Urs Baltensperger[1], and André S. H. Prévôt[1]

[1]Laboratory of Atmospheric Chemistry, Paul Scherrer Institute, 5232 Villigen PSI, Switzerland
[2]Laboratoire Inter-Universitaire des Systèmes Atmosphériques (LISA), UMR CNRS 7583, Université Paris Est Créteil et Université Paris Diderot, Institut Pierre Simon Laplace, Créteil, France
[3]School of Physics, Centre for Climate and Air Pollution Studies, Ryan Institute, National University of Ireland Galway, University Road, H91CF50 Galway, Ireland
[4]Marine and Renewable Energy Ireland
[5]Institute of Environmental Assessment and Water Research (IDAEA), CSIC, 08034 Barcelona, Spain
*Now at: Department of Chemical Engineering, Carnegie Mellon University, Pittsburgh, USA

*Correspondence to*: Sebnem Aksoyoglu (sebnem.aksoyoglu@psi.ch), Jianhui Jiang (jianhui.jiang@psi.ch)

**Abstract.** Biogenic volatile organic compound (BVOC) emissions are one of the essential inputs for chemical transport models (CTMs), but their estimates are associated with large uncertainties leading to significant influences on air quality modelling. This study aims at investigating the effects of using different BVOC emission models on the performance of a CTM in simulating secondary pollutants, i.e. ozone, organic and inorganic aerosols. The European air quality was simulated for the year 2011 by the regional air quality model Comprehensive Air Quality Model with Extensions (CAMx) version 6.3, using BVOC emissions calculated by two emission models: the Paul Scherrer Institute (PSI) model and the Model of Emissions of Gases and Aerosol from Nature (MEGAN) v2.1. Comparison of isoprene and monoterpene emissions from both models showed large differences in their general amounts as well as their spatial distribution both in summer and winter. MEGAN produced more isoprene emissions by a factor of 3 while the PSI model generated three times of monoterpene emissions in summer, while there was negligible difference (~4%) in sesquiterpene emissions associated with the two models. Despite the large differences in isoprene emissions (i.e. 3-fold), the resulting impact in predicted summer-time ozone proved to be minor (<10%, $O_3$-MEGAN was higher than $O_3$-PSI by ~7 ppb). Comparisons with measurements from the European air quality database (AirBase) indicated that PSI emissions might improve the model performance at low ozone concentrations, but worsen it at high ozone levels (>60 ppb). A much larger effect of the different BVOC emissions was found for the secondary organic aerosol (SOA) concentrations. The higher monoterpene emissions (a factor of ~3) by the PSI model led to higher SOA by ~110% on average in summer, compared to MEGAN, lead to better agreement between modelled and measured organic aerosol (OA): the mean bias between modelled and measured OA at 9 measurement stations using Aerodyne aerosol chemical speciation monitors (ACSM) / Aerodyne aerosol mass spectrometers (AMS) was reduced by 21% – 83% in rural/remote stations. Effects on inorganic aerosols (particulate nitrate, sulphate, and ammonia) were relatively smaller (< 15%).

## 1 Introduction

Biogenic volatile organic compounds (BVOCs) from the terrestrial biosphere play an important role in atmospheric chemistry. They affect production of ozone i(Calfapietra et al., 2013; Curci et al., 2009), formation process of secondary organic aerosol (SIA) (Aksoyoglu et al., 2017), and are the largest source of secondary organic aerosol (SOA) worldwide (Bonn et al., 2004; Hallquist et al., 2009; Hodzic et al., 2016; Kirkby et al., 2016). Emissions of BVOCs such as isoprene, monoterpenes (MT) and sesquiterpenes (SQT) are now commonly used as inputs within numerous chemical transport models (CTMs). While in many model inter-comparison studies anthropogenic emissions are harmonized, biogenic emissions usually differ (Bessagnet et al., 2016; Colette et al., 2017; Im et al., 2015; Solazzo et al., 2012). Different approaches in biogenic emission models may result in substantial difference on predicted emission rates of BVOCs (Messina et al., 2016; Oderbolz et al., 2013). Although there are a few studies comparing different BVOC models (Karl et al., 2009; Keenan et al., 2009; Steinbrecher et al., 2009), comprehensive studies showing the impacts of using different BVOC emission models on secondary pollutants in Europe are scarce. Some studies report the effect of biogenic emissions with zero-out simulations (Sartelet et al., 2012) or with doubled BVOC emissions (Aksoyoglu et al., 2017; Ciarelli et al., 2016). Curci et al. (2009) compared effects of two different biogenic emission inventories, one based on Guenther et al. (1995) and one based on Steinbrecher et al. (2009), on ozone in Europe during 1997 to 2003. However, the limitation of ozone production might have been altered due to large emission reductions of the various precursors in Europe during the past decades. Understanding the potential influence of biogenic emissions on European air quality is therefore of great importance especially under the continuously reduced anthropogenic emissions since early 1990s.

BVOCs, dominated by isoprene and monoterpenes, are generated from biosynthesis of precursor isopentenyl pyrophosphate in plants (Kesselmeier and Staudt, 1999). Isoprene is emitted from leaf surfaces immediately after synthesis (referred to as synthesis emission), while monoterpenes are mostly stored in plant organs after their production (pool emission) and some monoterpene species have synthesis emissions as well. The synthesis and emission processes could be influenced by various factors, such as plant species, foliage biomass, temperature, solar radiation as well as carbon and water availability (Grote and Niinemets, 2008), leading to high uncertainty in the estimates of BVOC emissions. Current BVOC emission models are mostly based on an empirical bottom-up approach by emission factors as a function of leaf temperature and photosynthetically active radiation (PAR) (Andreani-Aksoyoglu and Keller, 1995; Guenther et al., 2006; Guenther et al., 2012; Solmon et al., 2004). Although most of these models share similar algorithms, the inputs such as emission factors and land use types might vary widely for different studies. For example, the widely used MEGAN (Model of Emissions of Gases and Aerosols from Nature) (Guenther et al., 2012) estimates 19 categories of BVOC species by emission factors based on 15 CLM4 (Community Land Model) plant function types (PFT) (e.g., broadleaf evergreen tropical tree, broadleaf deciduous temperate shrub, etc.). To account for variability of different tree species within the same PFT, MEGAN2.1 provides emission factors for more than 2000 ecoregions worldwide based on tree species composition and tree-species-specific emission factors (Guenther et al., 2012). For regional simulations in which more detailed land use and vegetation information were available,

Solmon et al. (2004) estimated isoprene and monoterpenes emissions in France based on Corine Land Cover (CLC) land use data with a resolution of 50 – 100 m and BVOC emission factors of each tree species. Significant influences of land use and vegetation on the spatial distribution and magnitude of estimated BVOC emissions have been reported by many studies (Hantson et al., 2017; Oderbolz et al., 2013; Rosenkranz et al., 2015; Steinbrecher et al., 2009; Szogs et al., 2017).

5 As an important input to air quality models, BVOC emissions strongly influence simulated concentration of ozone and aerosols, with great spatial and temporal difference. BVOCs play crucial roles in both formation and removal processes of ozone (Calfapietra et al., 2013). Comparison between MEGAN and another widely used biogenic emission model, the Biogenic Emission Inventory System (BEIS) indicated that the influence of biogenic emission models on ozone simulation results over the United States is far greater than using a different photosynthetically active radiation (PAR) input (Zhang et al., 2017). The

10 potential influence of biogenic emissions on aerosol modelling results is more complicated. BVOCs are oxidized by reactions with oxidants like hydroxyl radicals (OH), nitrate radicals ($NO_3$) and ozone ($O_3$), and generate secondary organic aerosols via gas-to-particle partitioning (Griffin et al., 1999; Hoffmann et al., 1997). Different estimates of BVOC emissions directly influence the amount of biogenic SOA precursors (mainly MTs and SQTs) (Bonn et al., 2004), while they also influence indirectly the aerosol simulations via effects on oxidants (Ayres et al., 2015; Calfapietra et al., 2013; Ng et al., 2017).

15 Significant influence of anthropogenic activities and climate conditions on biogenic SOA formation (Carlton et al., 2010; Fu et al., 2014; Hoyle et al., 2011) makes it even more challenging to understand the effect of BVOC emissions on SOA simulations. Moreover, BVOCs also influence the secondary inorganic aerosol formation by changing the oxidant concentrations (Aksoyoglu et al., 2017; Karambelas, 2013; Sotiropoulou et al., 2004; Zhang et al., 2016). Aksoyoglu et al. (2017) found that doubled BVOC emissions in Europe led to an increase of particulate inorganic nitrate concentrations by up

20 to 35%.

 In spite of an increasing interest in understanding the influences of biogenic emissions on ozone and aerosols, limitations still remain: most of the studies focus on short periods (mostly in summer), while the potential influence of BVOC on SOA could still be high in winter at local scale, the evaluation of modelled OA is challenged by the scarcity of field measurements, and not much attention has been paid to the effects of BVOC on SIA by different biogenic models. In this study, we investigated

25 the effects of different estimates of BVOC emissions on modelled ozone and aerosol concentrations in Europe. Biogenic emissions were estimated by two BVOC emission models with different land cover inputs and emission factors: MEGAN as a widely used model globally and the PSI model to represent models developed for a specific region. The BVOC emissions from the two models were then used as input for the regional air quality model Comprehensive Air Quality Model with extensions (CAMx) to simulate gaseous and particulate pollutant concentrations in 2011. The modelled results were evaluated

30 by comparisons with ozone measurements from European air quality database AirBase and aerosol measurement from 9 Aerodyne aerosol chemical speciation monitor (ACSM)/Aerodyne aerosol mass spectrometer (AMS) stations over Europe.

## 2 Method and data

### 2.1 Regional air quality model CAMx

The regional air quality model CAMx version 6.3 (http://www.camx.com/) with the VBS (volatility basis set) scheme (Koo et al., 2014) was used to simulate the year 2011 in this study. The model domain (15ºW - 35ºE, 35ºN - 70ºN) covered Europe

with a horizontal resolution of 0.25º × 0.125º. The meteorological inputs were prepared by the Weather Research and Forecasting Model WRF-ARW (Advanced Research WRF) version 3.7.1 (NCAR, 2016; Skamarock et al., 2008). We used the ECMWF (European Centre for Medium-Range Weather Forecasts) global atmospheric reanalysis ERA-Interim data as initial and boundary conditions for the WRF model, with a spatial resolution of 0.72º × 0.72º and a time step of 6 hours (Dee et al., 2011). The meteorological fields from the WRF output were further processed by WRF-CAMx version 4.4

(http://www.camx.com/download/support-software.aspx) to match the CAMx vertical layers and to prepare the required parameters (e.g. vertical diffusivity). In CAMx, there were 14 terrain-following vertical layers reaching up to 460 hPa, with the first layer being ~20 m thick. The Carbon Bond 6 Revision 2 (CB6r2) mechanism (Hildebrandt Ruiz and Yarwood, 2013) was used for the gas-phase chemistry. Aqueous sulphate and nitrate formation in resolved cloud water was simulated by the RADM algorithm (Chang et al., 1987). Partitioning of inorganic aerosol components between the gas and particle phases was

calculated by the ISORROPIA thermodynamic model (Nenes et al., 1998). Organic aerosol formation from anthropogenic (including both land and ships) and biogenic (terrestrial) sources was modelled by the 1.5-D VBS organic aerosol chemistry/partitioning module (Koo et al., 2014), which describes the evolution of OA in the 2-D space of oxidation state and volatility. The standard CAMx v6.3 treats the aging and partitioning processes of secondary aerosols from biogenic and biomass burning sources in the same basis sets. To distinguish the contributions of biogenic and biomass burning sources to

OA, we separated the combined basis set VBS/PBS (V – Vapor, P – Particle, S – Secondary, B – Biogenic and Biomass Burning) into two sets: VBIS/PBIS (BI - Biogenic) for biogenic sources, and VBBS/PBBS (BB – Biomass Burning) for biomass burning sources.

The gridded initial concentrations of chemical species in each layer of the model domain, as well as at the domain lateral boundaries were obtained from the global model data MOZART-4/GEOS-5 (Horowitz et al., 2003) with a time resolution of

6 hours.  The ozone column densities were obtained from Total Ozone Mapping Spectrometer (TOMS) data by the National Aeronautics and Space Administration (ftp://toms.gsfc.nasa.gov/pub/omi/data/) and photolysis rates were calculated using the Tropospheric Ultraviolet and Visible (TUV) Radiation Model version 4.8 (NCAR, 2011). Anthropogenic emissions of non-methane volatile organic compounds (NMVOCs), $SO_2$, $NO_x$, CO, $NH_3$, $PM_{10}$ and $PM_{2.5}$ were obtained from the high-resolution European emission inventory TNO-MACC (Monitoring Atmospheric Composition and Climate)-III. As an update to TNO-

MACC-II (Kuenen et al., 2014), TNO-MACC-III has a major improvement in spatial distribution proxies, especially for urban areas (van Der Gon, 2015). The NMVOCs speciation was conducted following the approach described by Passant (2002). The PM emissions were further split into organic carbon, elemental carbon, sodium, sulphate and crustal minerals, based on country specific profiles provided by TNO.

## 2.2 Biogenic emission models

Two different biogenic emission models were used to calculate BVOCs emissions (isoprene, MT, SQT), i.e., MEGAN version 2.1 (Guenther et al., 2012) and the BVOC model developed by the Laboratory of Atmospheric Chemistry, Paul Scherrer Institute (Andreani-Aksoyoglu and Keller, 1995) (referred to as PSI model in this study). MEGAN is among the most widely used modelling systems estimating emission rates of BVOCs from terrestrial ecosystems. The MEGAN version 2.1 covers 147 individual BVOC species within 19 categories (Guenther et al., 2012). The PSI model was first developed for fine-resolution estimation of monoterpene and isoprene emissions in Switzerland (Andreani-Aksoyoglu and Keller, 1995) and was later expanded to the European domain (Oderbolz et al. (2013) Oikonomakis et al. (2018). Both the MEGAN and PSI model estimate biogenic emissions by an empirical bottom-up approach with similar algorithms based on standard emission rates (at leaf temperature of 30 °C and photosynthetically active radiation of 1000 µmol m$^{-2}$ s$^{-1}$) and the emission response to environmental conditions (Guenther et al., 2012;Andreani-Aksoyoglu and Keller, 1995). The major difference between the two models is that MEGAN uses PFT (plant function type) specific emission factors, while the PSI model uses plant species specific emission factors. Here we mainly focus on the differences in the calculation of emission rates and inputs of land use and vegetation. A general comparison between the major inputs of the PSI model and MEGAN v2.1 is presented in Table 1.

### 2.2.1 Emission rates

MEGAN estimates the reference emission rates by emission factors of 15 PFTs as listed in Table 1. The Global Emission Factors (version 2011) from the MEGAN website (http://lar.wsu.edu/megan/guides.html) were used in this study. Emission factors of compounds are given for each of the 15 PFTs (Guenther et al., 2012). Tree species based emission factors and forest species composition profiles for more than 2000 ecoregions worldwide were used to generate the high resolution (0.0083° × 0.0083°) global emission factor dataset. On the other hand, the PSI model uses reference emission rates of typical plant species in Europe (see Table 1). The reference emission rates (µg g(dry weight)$^{-1}$ h$^{-1}$) of isoprene and monoterpenes from forests, pasture and crops were calculated based on algorithms given by Lamb et al. (1993). Isoprene emissions from Norway spruce were assumed to be about 10% of α-pinene emission rates during daytime (Steinbrecher, 1989). Sesquiterpenes (SQT) are the least studied among the identified BVOCs due to their high reactivity and relatively low vapor pressure (Duhl et al., 2008). Determination of their basal emission rates is therefore challenging. In the PSI model, SQT emissions were treated only as pool emissions and assumed to be 5% (by weight) of the monoterpene emissions based on the emission rate data for 116 species compiled from various studies as given by Steinbrecher et al. (2009).

### 2.2.2 Response functions

Isoprene, one of the most important BVOC species, is released after biosynthesis by volatilization, which depends on both temperature and solar radiation. On the other hand, monoterpenes are stored in large storage pools after their production in the plant organs. Emissions of monoterpenes are mostly temperature-dependent, although there are some species that have both

light and temperature dependent synthesis emissions of MTs (Tingey et al., 1980). In the PSI model, the isoprene emissions are corrected by light ($\gamma_L$) and temperature ($\gamma_T$) response functions based on the algorithm described by Guenther et al. (1993):

$$\gamma_L = \frac{\alpha C_{L1} PAR}{\sqrt{1 + \alpha^2 PAR^2}} \tag{1}$$

$$\gamma_T = \frac{\exp[C_{T1}(T - T_s)/RT_sT]}{1 + \exp[C_{T2}(T - T_M)/RT_sT]} \tag{2}$$

where $\alpha$ (= 0.0027), $C_{L1}$ (= 1.066), $C_{T1}$ (= 95,000 J mol$^{-1}$), $C_{T2}$ (= 230,000 J mol$^{-1}$), $T_M$ (= 314 K) are all empirical coefficients determined by nonlinear fitting based on emission rate measurements, $R$ is the gas constant (8.314 J K$^{-1}$ mol$^{-1}$), and $T_s$ is the standard leaf temperature (303.16 K). The response functions of the isoprene emission in MEGAN are based on Guenther et al. (1999), an updated version of Guenther et al. (1993). The major difference of the improved algorithm is the inclusion of the influence of past temperature and light conditions. New empirical coefficients $T_{opt}$ and $E_{opt}$ calculated by the average leaf temperature over the past 24 and 240 hours are added to include the continuous influence over time, respectively (Eq. (3)):

$$\gamma_T = \frac{E_{opt} \times C_{T2} \times \exp(C_{T1} \times x)}{C_{T2} - C_{T1} \times (1 - \exp(C_{T2} \times x))} \tag{3}$$

where $x = [(1/T_{opt}) - (1/T)]/0.00831$. A detailed introduction to $T_{opt}$ and $E_{opt}$ can be found in Guenther et al. (2006).

For the light-independent response of MT pool emissions, similar exponential corrections are used by MEGAN and the PSI model which are based on Lamb et al. (1993) and Tingey et al. (1980) as shown in Eq. (4):

$$E = E_s \times \exp(\beta \times (T - Ts)) \tag{4}$$

where $E$ is the MT emission at temperature $T$, $E_s$ is the emission under standard conditions ($T_s$ = 30 ºC), and $\beta$ is the slope coefficient of dlnE dT$^{-1}$. The slope value $\beta$ has a wide range between 0.057 to 0.144 according to previous literature (Guenther et al., 1993). The value of 0.1 is used for most MT species (e.g. α-pinene, β-pinene, 3-carene and limonene) in MEGAN2.1, while the values are between 0.065 to 0.077 for different MT species in the PSI model. The light-dependent synthesis emissions of MTs were considered in MEGAN v2.1 as described in Guenther et al. (2012). Depending on different MT species, the light-dependent fraction of MT emissions ranges between 0.2 to 0.8 for MEGAN. In the PSI model, the light-dependent emissions from Norway spruce are calculated for each monoterpene species as a function of PAR based on the data of Schürmann (1993). In addition to the light and temperature response, MEGAN v2.1 covers also some other factors such as leaf age and leaf area index (Guenther et al., 2012). Since the correction of soil moisture and $CO_2$ dependence are not included in the offline version of MEGAN (Emmerson et al., 2016), we used the default parameterization where the correction factors were set to 1.

The variation of light and temperature within the forest canopy are corrected by a canopy model in both the PSI model and MEGAN. The PSI model uses the canopy model by Baldocchi et al. (1985) combined with experiments in Hartheim Forest (Germany) and Central Switzerland (Joss, 1995). The detailed algorithm of the canopy correction for the PSI model was reported by Keller et al. (1995). The MEGAN canopy environmental model is based on Guenther et al. (1999), which estimates

incident PAR and temperature of sun and shade leaves at different canopy depths. Details can be found in Guenther et al. (2006; 2012). A BVOC reduction of about ~20% due to the canopy model was reported for the PSI model by Oderbolz et al. (2013). Although different canopy models could influence the modelled BVOC emission, such influence was within the uncertainty range of observed fluxes (Guenther et al., 2006; Lamb et al., 1996).

### 2.2.3 Inputs of driving variables

Three types of basic driving variables are required for both MEGAN and the PSI model, namely meteorological conditions, land use and biomass density. The meteorological data provide hourly, gridded information of temperature, solar radiation, wind speed, moisture and surface pressure to drive the model simulation of emission response. We used the same meteorological data retrieved from the WRF-ARW model as input for both models. The main difference between the two model inputs is in the land use and leaf biomass density.

MEGAN v2.1 uses the Community Land Model version 4 (CLM4) including 15 PFTs as shown in Table 1. In this study, we adopted for MEGAN the same global PFT map as in Sindelarova et al. (2014) with a resolution of 0.05 degrees. For the PSI model, the GlobCover 2006 inventory by European Space Agency (http://due.esrin.esa.int/page_globcover.php) was used. This inventory was developed based on MERIS (MEdium Resolution Imaging Spectrometer) FRS (Fine Resolution Full Swath product) level 1B data during December 2004 to June 2006 (Bicheron et al., 2008). The raw data has a fine resolution of 300 m and 64 categories of land use types e.g. needleleaved evergreen forest, broadleaved deciduous forest, and mixed broadleaved and needleleaved forest. The grid-scale fractions of needleleaf, broadleaf and mixed forests were first calculated based on the GlobCover inventory data. The mixed forest was assumed to be composed of 50% needleleaf and 50% broadleaf species. Different tree species in the same category may have different emission factors. For instance, although all belong to broadleaf tree species, oak (*Quercus*) has high emission rate, while beech (*Fagus sylvatica*) and maple (*Acer*) are negligible BVOC emitters. Even within the same genus, there might be large differences in emissions, e.g. two oak species, where quercus robur is a high isoprene emitter and quercus suber a low isoprene emitter (Steinbrecher et al., 2009). Europe has relatively lower abundance of flora in both diversity and numbers, and 6 tree species cover 2/3 of the forest area, namely namely Scots pine, Norway spruce, beech, maritime pine, European oak and evergreen oak (Simpson et al., 1999). Therefore, in the PSI model, we classified the forests into 10 typical forest species (see Table 1 Vegetation Class 1–10) found in Europe based on the country-specific forest species profile from Simpson et al. (1999). The original 35 forest species in Simpson et al. (1999) were grouped into 10 classes (including 5 coniferous species and 5 broadleaf species), and the ratio of each species class to the total coniferous forest and broadleaf forest was calculated (Table S2). The ratio of "other trees" were proportionally added to the 10 tree species. As the "other trees" are mainly in a few Mediterranean countries, their influences on the whole domain is small. The species-coverage was then generated by multiplying the forest coverage from GlobCover with the country-specific tree species profile.

The biomass density in MEGAN was calculated by the canopy environment module based on the satellite data of the leaf area index (LAI, m2 leaves per m2 projected area) with a time step of 8 days. The TERRA/MODIS vegetation data products

MOD15A2 were downloaded from the NASA Earth Observations website (https://neo.sci.gsfc.nasa.gov/view.php?datasetId=MOD15A2_E_LAI&year=2011). The grid-scale LAI was then divided by the fraction of vegetation coverage of each grid (sum of PFT) to get the average LAI of vegetation covered surfaces (LAIv).

As the reference emission rates of the PSI model are based on dry weight of leaf biomass, the leaf biomass density factors (g dry weight per $m^2$ projected area) of each tree species (Cannell, 1982; Satoo and Madgwick, 1982) were explicitly used in the PSI model. To simulate the vertical variation of foliar biomass in the canopy, the biomass density was scaled by the leaf area distribution in each canopy layer as described in Oderbolz et al. (2013). The temporal variation of the biomass was simulated by monthly factors for different plant types. For example, the PSI model assumes that the leaf biomass of deciduous trees, such as oak and larch, turn to zero in the winter months (November - March) and crops only have biomass in the growing season (April - August).

## 2.3 Observation datasets and statistics

Two types of measurement datasets were used to evaluate the model results. Measurements of hourly ozone concentrations in 2011 were extracted from the European air quality database AirBase v7 by the European Environment Agency (Mol and Leeuw, 2005). To reduce the uncertainties arising from the model resolution, only ozone measurements at background-rural stations were used in the model evaluation. Concentrations of OA and secondary inorganic aerosol (particulate nitrate, sulphate, and ammonium) were obtained from ACSM/ AMS measurements at 9 stations: Zurich (Canonaco et al., 2013), Mace Head (Ovadnevaite et al., 2014; Schmale et al., 2017), Montsec (Ripoll et al., 2015), Bologna and San Pietro Capofiume (Gilardoni et al., 2014), Paris SIRTA (Site Instrumental de Recherche par Télédétection Atmosphérique) (Petit et al., 2015), Marseille (Bozzetti et al., 2017), Finokalia (as continuation of Hildebrandt et al. (2010)), and the SMEAR II (Station for Measuring Forest Ecosystem–Aerosol Relations) Hyytiälä station (Kortelainen et al., 2017). The spatial distribution of the measurement sites is shown in Fig. 1. Zurich, Bologna and Marseille are urban sites, Paris SIRTA is a suburban site, while Mace Head, Finokalia, San Pietro Capofiume and Montsec are in rural or remote areas. We divided the whole domain into 3 regions to enable a comparison for different latitudes: northern Europe (NE, 55°N – 70°N), central Europe (CE, 45°N – 55°N) and southern Europe (SE, 35°N – 45°N). The time span of OA observations at each station is shown in Fig. S1. The measurements cover nearly the whole year of 2011 in Zurich (except for January) and Mace Head (except for November and December), while other stations cover shorter periods (Fig. S1). The modelled concentrations at the surface (1st layer) were interpolated to the location of the stations to compare with the measurements. The statistical metrics, such as mean bias (MB), mean error (ME), mean fractional bias (MFB), mean fractional error (MFE) and root mean square error (RMSE), were calculated and compared for two CAMx simulations using different BVOC emissions obtained by MEGAN and the PSI model. The definitions of these statistical metrics are presented in Table S1.

# 3 Results and discussion

## 3.1 Biogenic VOCs in Europe

BVOC emissions estimated by the PSI model and MEGAN showed significant differences in both spatial and temporal variations. To evaluate the seasonal differences, we compared the BVOC emissions in February and July to represent winter and summer periods, respectively. BVOC emissions in winter are much lower than in summer, especially for isoprene which is mainly emitted by deciduous broadleaf trees. The PSI model produced negligible isoprene in winter, as the leaf biomass of oak trees, the largest isoprene emitters, was set to zero during that period. For monoterpenes, which are mainly emitted by evergreen needleleaf forests, the seasonal difference was less obvious than for isoprene, although the emissions in winter were lower than in summer due to lower temperatures (about 82% and 96% lower than in summer for the PSI model and for MEGAN, respectively).

Isoprene emissions by MEGAN were substantially higher than those in the PSI model (Fig. 2a) by a factor of 2.9 on average in summer. The highest difference occurred in Southern Europe (Fig. S2a), where the highest grid-scale absolute difference (MEGAN – PSI) reached 203 kg cell$^{-1}$ h$^{-1}$ in Spain. The major reason for low isoprene emissions by the PSI model is the assumption of oak being the main broadleaf tree species emitting isoprene, while all the broadleaf trees and shrubs (PFT4 - PFT11) have positive emission factors in MEGAN. On the other hand, the PSI model estimated in general more monoterpene emissions than MEGAN (Fig. 2b). The total emissions in the whole domain were 486 t h$^{-1}$ (winter) and 2768 t h$^{-1}$ (summer) for the PSI model, while the values were only 40 t h$^{-1}$ (winter) and 994 t h$^{-1}$ (summer) for MEGAN. Accordingly, the average MT emissions in the PSI model were higher than MEGAN by a factor of 12.2 and 2.8 in winter and summer, respectively. Significantly higher MT emissions by the PSI model can be found in Scandinavia, the Iberian Peninsula and southeast Europe (Fig. S2b). The only areas where the PSI model estimated lower MT emissions than MEGAN were in Italy, the Balkans and France, due to a relatively lower needleleaf forest coverage in these regions (Fig. S3). The difference in SQT emissions by two models was smaller in magnitude (average SQT-PSI is 4.1% higher than SQT-MEGAN in summer) compared with other BVOCs species with a similar pattern of spatial difference as for MT. (Fig 2c, Fig S2c).

The diurnal variations of the isoprene and monoterpene emissions showed a peak around noon for both models (Fig. 3). In winter, highest isoprene emissions occurred in Central Europe (CE) for PSI model, while in Southern Europe (SE) for MEGAN (Fig. 3a). The main reason is isoprene-PSI mainly came from Norway spruce in the CE instead of deciduous trees in the south during winter time. Comparison of the monoterpenes emissions (Fig. 3b) with temperature and photosynthetically active radiation (PAR) (Fig. 3c) indicates that monoterpenes emissions by the PSI model are mostly temperature-dependent while the influence of light is stronger for the MEGAN–MT emissions. For instance, the highest PSI–MT emissions in summer occurred at the same time of the highest temperature (13:00–14:00 UTC), while the occurrence of highest MEGAN–MT is close to the PAR peak (10:00–12:00 UTC). MEGAN showed steeper changes ([emission(t) - emission(t-1)] / emission(t)) due to larger slope coefficient $\beta$ value used in the exponential temperature response function, as well as potentially higher fraction of light-dependent MT emissions. Especially for monoterpenes in southern Europe (SE) in summer, the highest increase and

decrease rates reached 43.8% (at UTC 05:00) and -57.1% (at UTC 18:00), respectively, while in the PSI model the hourly changes varied between 18.6% (at UTC 9:00) and -15.6% (at UTC 19:00).

BVOC measurements are rare and the concentrations are associated with very high spatial gradients (especially vertical) due to high reactivity and local mixing processes that are unlikely captured by the model in the respective grid cell. Nevertheless but with these caveats in mind we compared a few measurements available for isoprene with our model results to get an idea about the range of differences. Compared to monoterpenes, there were more isoprene measurements at various European sites in 2011 (see Fig. 4). Clearly, the MEGAN-isoprene data are much higher than measurements at all 12 sites while the PSI-isoprene results are closer to the measurements.

Unlike the single compound of isoprene, monoterpenes consist of several species and therefore it is even more difficult to perform comparisons with measurements, which are rare and have large uncertainties. Only a limited number of MT measurements were reported in Europe (only in Finland) during our simulation period (Hakola et al., 2012; Hellen et al., 2012). Hakola et al. (2012) reported average MT concentrations of about 508 ppt (with a range between about 150 and 800 ppt) in August 2011 at SMEAR II station at Hyytiälä. MEGAN-MT for the same period was 117 ppt while PSI-MT was around 2 ppb (for the same site Rinne et al. (2005) reported MT concentrations of between 200-500 ppt during daytime and more than 1 ppb at nighttime in summer 2004). On the other hand, the measured MT concentrations at a nearby urban background station SMEARIII in Helsinki were lower, with around 117 ppt in summer (Hellen et al., 2012). Both models predicted higher concentrations for that site (MEGAN-MT 303 ppt, PSI-MT 1 ppb). In order to get an idea about the model performance in other regions, we compared our results also with MT concentrations measured at Hohenpeissenberg (southern Germany) in June 2006 (Oderbolz et al., 2013). Both model results (PSI-MT: 75 ppt, MEGAN-MT: 130 ppt) in that region were similar to measurements (~100 ppt). Although this comparison of measurements and model results for different years under different meteorological conditions has a very high uncertainty, it might help to understand the range of differences between the model results and the measurements. In general, all these comparisons suggest that MT concentrations might be underestimated using MEGAN emissions over Scandinavia while PSI emissions might be too high. On the other hand, both models seem to predict MT emissions relatively well in central Europe.

These results generally agree with previous inter-comparison studies. Studies comparing different models with each other, as well as with measurements suggest that MEGAN tends to overestimate isoprene emissions especially in Scandinavian countries and south-west Europe and to underestimate monoterpene emissions by more than a factor of 2 (Bash et al., 2016; Carlton and Baker, 2011; Emmerson et al., 2016; Poupkou et al., 2010; Silibello et al., 2017). However, due to limited measurement data and large uncertainties especially due to representativeness of measurement and modelled locations, it is not possible to conclude which model predicts more reliable BVOC emissions.

## 3.2 Influence of different BVOC emissions on the modelling of ozone and aerosols

### 3.2.1 Ozone

The modelled ozone mixing ratios from two simulations using the biogenic emissions calculated by the PSI model ($O_3$-PSI) and MEGAN ($O_3$-MEGAN) were evaluated by the measurements from the European air quality database AirBase (Mol and Leeuw, 2005). Table 2 shows the statistical metrics of modelled average mixing ratios of afternoon (12:00–18:00 UTC) surface ozone at 537 rural background stations. The model performance in summer was generally better than in winter for all regions, but the difference between the PSI model and MEGAN was small. In winter, the two models showed similar mean bias (~3 ppb) and RMSE (~9.2 ppb) between modelled and measured concentration. In summer, the PSI model showed lower (34.0%) mean bias but slightly higher (1.3%) RMSE than MEGAN. To investigate the difference in more detail, we compared the bias between modelled and observed $O_3$ in different mixing ratio bins for different regions in summer (Fig. 4). In general, ozone modelled using the BVOC emission input from both models was overestimated at low mixing ratios and underestimated at high mixing ratios. A similar pattern was found in previous $O_3$ modelling studies in Europe (Im et al., 2015; Oikonomakis et al., 2018; Solazzo et al., 2017). CAMx performed better with MEGAN emissions at most stations at the high ozone bins. Although the PSI model led to lower overall MB (Table 2), it was mostly due to compensation at the low and high $O_3$ level bins.

To further explore the reasons for the different model performance in the ozone simulations, we present the spatial distributions of modelled ozone in summer calculated using BVOC emissions from the PSI model and MEGAN in Fig. 5. $O_3$-PSI was generally lower than $O_3$-MEGAN in whole Europe. In summer, the largest effect of using different BVOC emissions on ozone was mostly in southern Europe, especially in the Mediterranean region, with the highest relative difference between $O_3$-PSI and $O_3$-MEGAN reaching -14% (7.5ppb, in Italy); while in UK and Ireland, where isoprene emissions by PSI model were higher than MEGAN (Fig. S2), a positive difference up to 3.9 ppb was found. The spatial distribution of the ozone difference, i.e. (PSI-$O_3$) – (MEGAN-$O_3$) (Fig. 6, right panel) is very similar to that of the difference in the isoprene emissions (Fig. S2a). As an important ozone precursor, isoprene reacts with hydroxyl radicals (OH) to form peroxyl radicals ($RO_2$, $HO_2$) which further react with NO to generate $NO_2$ and finally ozone (Wennberg et al., 2018). This process can be significantly affected by the availability of isoprene and $NO_x$ in the atmosphere as well as temperature (Calfapietra et al., 2013), leading to high uncertainties in the net influence of BVOC emissions. Li et al. (2007) found that increasing the isoprene emissions by 50% resulted in an increase of the $O_3$ mixing ratios by 5 –25 ppb in urban Houston in the United States, and Zare et al. (2012) suggested that the 21% higher annual isoprene emissions by MEGAN than GEIA (Global Emissions Inventory Activity) led up to 10% higher $O_3$ concentrations in the African Savannah. However, the effect of the BVOC emissions on the ozone levels in Europe was much smaller in this study. The about 3 times higher isoprene emissions in MEGAN led only to up to ~10% (7 ppb) higher ozone mixing ratios in summer compared to the PSI model. Similarly, an earlier study by Aksoyoglu et al. (2012) using the PSI model for BVOC emissions suggested that increasing the isoprene emissions by a factor of four in Europe led to an increase of less than 10% in the afternoon ozone mixing ratios. The main reason for the weak effect of the

isoprene emissions on ozone is the stronger sensitivity of ozone formation in general to $NO_x$ emissions rather than VOC emissions in Europe. An additional reason might be the rather low ozone production compared to the background ozone where the latter is not affected by local European emissions (Oikonomakis et al., 2018; Sartelet et al., 2012). Several European studies reported that ozone formation in most regions is $NO_x$-sensitive except around the English Channel, Benelux and Po Valley regions, where $NO_x$ emissions are high (due to intensive anthropogenic $NO_x$ emissions from both land and shipping or geographical characteristics leading to high accumulation of pollutants) and the response to a change in the VOC emissions is relatively stronger (Aksoyoglu et al., 2012; Beekmann and Vautard, 2010; Oikonomakis et al., 2018). However, the sensitivity of ozone formation to its precursor emissions might be changing as a result of large $NO_x$ emission reductions in Europe since 1990 according to the Gothenburg Protocol. On the other hand, emissions from shipping activities are not regulated as strictly as land emissions and have been increasing continuously especially in the Mediterranean, affecting both ozone and particulate matter concentrations (Aksoyoglu et al., 2016; Viana et al., 2014).

### 3.2.2 Organic aerosols

The effects of different BVOC emissions on organic aerosols were investigated by comparing modelled OA concentrations with measurements at 9 ACSM/AMS stations. Although the OA concentrations were generally underpredicted in both cases, the model performance for OA was better with the PSI biogenic emissions (Fig. 7). About 67% of the modelled OA concentrations were below the 1:2 line in the case of MEGAN (Fig. 7b). The mean bias between observed and modelled OA concentrations with the PSI BVOC emissions was lower than the bias obtained with MEGAN emissions (3.9% in Paris – 83.4% in Mace Head, see Table 3). The better model performance when using the PSI emissions was more obvious at rural or remote stations where biogenic sources play a major role in OA formation. The mean bias of OA by the PSI model was 21 % to 83% lower than MEGAN at rural or remote stations (Finokalia, San Pietro Capofiume, Montsec, SMEAR II and Mace Head), while the range was 4% - 12% for Paris, Bologna and Marseille (see Table 3). The situation of Zurich was different with an MB reduction of 67% by PSI model compared with MEGAN as an urban station, mostly because the station is an urban background site that is strongly affected by biogenic emissions (Daellenbach et al., 2017).

We further evaluated the model performance of the temporal variation at Zurich and Mace Head as examples of urban background and rural stations, respectively (Fig. 8, top panels), because these two datasets covered almost the whole year. In spite of some underestimation, the temporal variation was well captured. At Zurich, the difference between the two cases (OA-PSI and OA-MEGAN) was small in February and March and they were both lower than the measurements, possibly due to underestimation of biomass burning OA (Fountoukis et al., 2014). The largest difference occurred in the fall when OA-PSI reproduced the measurements quite well, while OA-MEGAN showed a large underestimation. This is consistent with source apportionment studies performed for Zurich (Canonaco et al., 2013; Canonaco et al., In prep.; Daellenbach et al., 2017), which reported that the contribution of biogenic sources to OA was minor in the period of January to March but significant (>50%) in summer and fall.

The situation was quite different for Mace Head. Located on the west coast of Ireland and 90 km away from the closest city Galway (Schmale et al., 2017), Mace Head is a remote station with low influence from the anthropogenic activities (O'Dowd et al., 2014). The simulation with the PSI biogenic emission model could reproduce all the measured peaks quite well, while the simulation using the MEGAN emissions failed to capture their magnitude. To investigate the cause of the high OA concentrations during certain periods, 72-hour back trajectory analyses ending at Mace Head on 26 March (as an example for a high-OA day) and on 4 August (as an example for a low-OA day) were conducted by NOAA's HYSPLIT atmospheric transport and dispersion modelling system (Stein et al., 2015). According to the HYSPLIT results (Fig. S4), the air masses were transported from Ireland and Scotland during the high-OA period (Fig. S4a), while during the low-OA period the air masses came from the North Atlantic Ocean (Fig. S4b), suggesting that the OA peaks originated from anthropogenic or biogenic sources on land. The influence of wind direction was further studied by comparing modelled and measured OA during the two periods featured by land-wind (24-26 March) and marine-wind (2-4 August) in Fig. S5. Measured OA in period with dominant wind direction from land was higher than during the marine-wind dominant periods by a factor of ~10. Modelled OA-PSI was very close to measurements while OA-MEGAN was underestimated in both periods. However, it is not possible to conclude that the good model performance for OA with PSI emissions is due to the fact that its high MT emissions are more accurate. It could also be due to the overestimated MT emissions compensating other missing continental sources of OA, e.g. biomass burning.

The spatial distribution of the SOA difference showed a similar pattern as its main precursor, monoterpenes (Fig. 9). The PSI emissions lead to significantly higher (by 113% and 109% in winter and summer, respectively) SOA production than MEGAN. The grid-scale difference reached up to a factor of 35 and 17 for winter and summer SOA, respectively. The largest differences occurred in central Europe, the Iberian Peninsula and Turkey in winter, and especially in Scandinavia in summer.

The modelled POA was also slightly higher (6.5% in winter and 7.8% in summer for average) with PSI emissions compared to the case with MEGAN (Fig. S6). Unlike in the traditional CTMs, where POA is treated as inert, the VBS scheme of CAMx allows POA to evaporate and react with oxidants. According to the partitioning theory (Donahue et al., 2006; Odum et al., 1996), higher total OA concentrations lead to higher partitioning to the particle phase for all compounds that are soluble in the aerosol matrix. Therefore, in our case, the high OA-PSI shifted the particle – gas equilibrium of primary condensable gases towards the particle phase, resulting in higher POA.

### 3.2.3 Inorganic aerosols

The influence of BVOC emissions on secondary inorganic aerosols (SIA) was much smaller than on SOA according to the comparison of model results with measurements (Table 3). At the nine ACSM/AMS stations, using the PSI emissions generally reduced the RMSE between modelled and measured particulate nitrate ($PNO_3$), sulphate ($PSO_4$), and ammonium ($PNH_4$) by up to 15.0%, 1.7%, and 7.7%, respectively, compared to CAMx simulations with the MEGAN emissions. Only Finokalia (Greece, rural) and San Pietro Capofiume (Italy, rural) had lower RMSE with the MEGAN emissions. Unlike the obvious difference in OA, difference between the modelled temporal variations of the inorganic aerosol was negligible with the two

emission estimates (Fig. 8). The $PNO_3$-MEGAN was slightly higher than PSI, because lower MT emission by MEGAN lead to lower $MT$-$NO_3$ reaction and therefore more $NO_x$ were available to be oxidized to $PNO_3$ (Fig. S8).

The modelled and measured daily average concentrations match well except for February and March at Zurich, when temperature was significantly underestimated and resulted in higher condensation (Fig. S8a). A similar effect of temperature was not observed in OA during the same period, possibly due to compensation of underestimated winter OA as a consequence of lacking sources in the model, especially biomass burning (Ciarelli et al., 2017). On the other hand, the modelled primary elemental carbon (PEC) matched the measurements at Zurich very well.

Similar to the situation of OA, the measured SIA ($PSO_4$, $PNO_3$ and $PNH_4$) at Mace Head peaked during the periods with wind from land. Both biogenic models captured the peaks well but overestimated the SIA during the peak periods. The modelled elemental carbon concentrations (PEC, in Fig. 8) on the other hand, were lower than the measured equivalent black carbon (EBC) in general but followed the temporal variation very well. In a study about the aerosols at Mace Head, O'Dowd et al. (2014) reported that EBC measurements can significantly overestimate black carbon concentration by up to 50% or more. Overestimation of SIA could result from either too high precursor emissions or too much particle formation in the aqueous phase. The precursor gases $SO_2$ and $NO_x$ from anthropogenic sources (continental, shipping) (Fig. S8) might be accumulated too much in the surface layer since all emissions were injected into the $1^{st}$ layer, leading to too high SIA formation.

The differences in the spatial distributions of $PSO_4$, $PNO_3$ and $PNH_4$ between the two simulations with PSI and MEGAN emissions are shown in Fig. 10. The inorganic aerosol concentrations varied by less than 15% on the grid scale for the different BVOC emissions. Highest $PSO_4$ levels were predicted in central and eastern Europe in winter, where $SO_2$ emissions are higher, while in summer the elevated sulphate concentrations were mostly along the shipping routes (Fig. 9). The PSI BVOC emissions lead to higher $PSO_4$ than MEGAN, especially over the area from southern Poland to Turkey through the Balkan Peninsula in summer. These regions have the highest $SO_2$ emissions in the model domain due to large combustion based power plants and coal burning. In summer, the main pathway for sulphate formation in southern Europe is the gas-phase oxidation of $SO_2$ with OH radical (Chrit et al., 2018; Megaritis et al., 2013). The higher sulphate concentrations predicted by CAMx with PSI BVOC emissions are consistent with the spatial pattern of the differences between PSI-MEGAN simulations for $SO_2$ concentrations and OH radicals (Fig. S9) due to the following: As reaction with OH radical is the largest loss pathway for isoprene in the atmosphere (Wennberg et al., 2018), higher isoprene emissions in MEGAN consumes more OH radical. As a consequence, less $SO_2$ is oxidized to form $PSO_4$ when MEGAN emissions are used (Fig. S9), leading to lower $PSO_4$ formation.

Formation of $PNO_3$ depends on the availability of $NO_x$ and $NH_3$ emissions (Aksoyoglu et al., 2011; Wen et al., 2015). In contrast to $PSO_4$, $PNO_3$ and $PNH_4$ concentrations modelled using the PSI biogenic emissions were generally lower than those using MEGAN emissions, especially in regions where PSI model has more MT emissions (Fig. S2b). Nitrate radicals are recognized as a significant sink for BVOCs, especially monoterpenes at night (while OH oxidation is more relevant for isoprene during daytime) (Kiendler-Scharr et al., 2016; Ng et al., 2017). Higher monoterpene emissions produced by the PSI model lead to larger consumption of nitrate radicals affecting $PNO_3$ formation from $HNO_3$ and $NH_3$. These results are consistent with a recent study showing the significant effect of BVOCs on ammonium nitrate (Aksoyoglu et al., 2017).

## 4 Conclusions

In this study, the European air quality in the year 2011 was simulated by the regional air quality model CAMx using two biogenic volatile organic compound (BVOC) emission models: MEGAN and PSI model. The model results were evaluated by $O_3$ measurements from the European air quality database AirBase v7 as well as the aerosol measurements at 8 ACSM/AMS stations. The results indicate that MEGAN generates more isoprene (by a factor of about 3), but less (~36%) monoterpene emissions than the PSI model in Europe in summer, mainly due to their different vegetation classification and reference emission rates. In spite of much higher isoprene emissions, simulations with MEGAN led to only slightly higher (7 ppb, <10%) ozone concentrations in summer compared to PSI emissions, especially in southern Europe. The evaluation of model results showed that PSI emissions improve the model performance for low ozone mixing ratios, but they worsen it at mixing ratios above 60 ppb.

The largest effect of using different BVOC emissions was predicted to be on SOA. PSI emissions led to higher SOA concentrations by about 110% compared to MEGAN due to higher monoterpene emissions, and therefore show a better model performance for OA at all 9 measurement sites. A more detailed evaluation of modelled organic and inorganic aerosols was performed at Zurich and Mace Head where aerosol measurements were available for relatively longer periods. Comparison of modelled and measured OA at Zurich suggested that OA concentrations could be captured very well with PSI BVOC emissions most of the time except in winter when modelled OA was underestimated by both PSI and MEGAN emissions. These results pointed out the missing winter sources such as biomass burning. On the other hand, at the remote site Mace Head, aerosol concentrations were affected by the prevailing air masses. Using PSI biogenic emissions, we could reproduce the OA peaks almost perfectly while OA concentrations were significantly underestimated when MEGAN biogenic emissions were used. One should however keep in mind that good model performance could also be due to compensation of other factors.

Effects of using different BVOC emission models on secondary inorganic aerosols (particulate nitrate, sulphate, ammonium) were relatively small (< 15%). The mean bias between modelled and measured values was lower when PSI model was used. The results of this study emphasize the importance of BVOC emissions in ozone and organic aerosol simulations and model inter-comparison studies. In future studies, emission factors should be improved in BVOC models to include more regional specific vegetation types to reduce the uncertainties in BVOC emission estimates and to improve air quality modeling results.

*Data availability.* The data of this study are available upon request from the corresponding authors.

*Author Contribution.* JJ and SA conceived the study. JJ carried out the model simulation and data analysis. GC and EO contributed to model setup. IEH, FC, COD, JO, MCM provided the measurement data and contributed to data interpretation. SA, ASHP and UB supervised the entire work development. The paper was prepared by JJ. All authors discussed and contributed to the final paper.

*Competing interests.* The authors declare that they have no conflict of interest.

*Acknowledgements.* We would like to thank the TNO for providing anthropogenic emissions, European Centre for Medium-Range Weather Forecasts (ECMWF) for the access to the meteorological data, the European Environmental Agency (EEA) for the air quality data, the National Aeronautics and Space Administration (NASA) and its data-contributing agencies (NCAR, UCAR) for the TOMS and MODIS data, the global air quality model data and the TUV model. Simulation of WRF and CAMx models were performed at the Swiss National Supercomputing Centre (CSCS). We thank the EBAS database by Norwegian Institute for Air Research (NILU) for the measurements data of isoprene concentration. We are grateful to RAMBOLL for the valuable support for CAMx. We thank the ACSM/AMS data providers, namely Stefania Gilardoni for Bologna and San Pietro Capofiume stations; Nicolas Marchand and MASSALYA Instrumental Platform (https://lce.univ-amu.fr/en/massalya) for Marseille; Olivier Favez (INERIS) and the whole SIRTA team for measurements conducted in the Paris area in the frame of the EU FP7 ACTRIS program under the grant agreement n° 262254; Kalliopi Florou for Finokalia; Liqing Hao and Annele Virtanen for SMEAR II Hyytiälä. EPA-Ireland (AEROSOURCE, 2016-CCRP-MS-31) is acknowledged, as well as EGAR group from IDAEA-CSIC (special mention to Anna Ripoll and Andrés Alastuey) and Generalitat de Catalunya (AGAUR 2017 SGR41). M.C. Minguillón acknowledges the Ramón y Cajal fellowship awarded by the Spanish Ministry of Economy, Industry and Competitiveness.

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

**Table 1.** Comparison between major input of PSI model and MEGAN v2.1

| Inputs | PSI Model | MEGAN v2.1 |
|---|---|---|
| Meteorology | WRF-ARW v3.7.1 | WRF-ARW v3.7.1 |
| Land-use | GlobCover 2006 inventory ($0.00028\,^{\circ} \times 0.00028\,^{\circ}$) | Community Land Model version 4 (CLM4, $0.05^{\circ} \times 0.05^{\circ}$) |
| | Vegetation class | Plant Functional Type (PFT) |
| | 1. Norway spruce (*Picea abies*) | 1. Needleleaf Evergreen Temperate Tree |
| | 2. Silver fir (*Abies alba*) | 2. Needleleaf Evergreen Boreal Tree |
| | 3. Scots pine (*Pinus sylvestris*) | 3. Needleleaf Deciduous Boreal Tree |
| | 4. Arolla pine (*Pinus cembra*) | 4. Broadleaf Evergreen Tropical Tree |
| | 5. European larch (*Larix decidua*) | 5. Broadleaf Evergreen Temperate Tree |
| | 6. European beech (*Fagus sylvatica*) | 6. Broadleaf Deciduous Tropical Tree |
| | 7. Sycamore maple (*Acer pseudoplanatus*) | 7. Broadleaf Deciduous Temperate Tree |
| | 8. Common ash (*Fraxinus excelcior*) | 8. Broadleaf Deciduous Boreal Tree |
| | 9. European oak (*Quercus robur*) | 9. Broadleaf Evergreen Temperate Shrub |
| | 10. Sweet chestnut (*Castanea sativa*) | 10. Broadleaf Deciduous Temperate Shrub |
| | 11. Pasture | 11. Broadleaf Deciduous Boreal Shrub |
| | 12. Crop | 12. Arctic C3 Grass |
| | | 13. Cool C3 Grass |
| | | 14. Warm C4 Grass |
| | | 15. Crop |
| Emission factors | Reference emission rate calculated based on Steinbrecher et al (2009) (Unit: $\mu g\ g_{dw}^{-1}\ h^{-1}$)[1] | Global Emission Factors Version 2011 from MEGAN website (Unit: $\mu g\ m^{-2}\ h^{-1}$) |
| Biomass density | Leaf biomass density (g dry weight per $m^2$ projected area) of each tree species obtained from Cannell (1982) and Satoo and Madgwick (1982) | TERRA/ MODIS (Moderate Resolution Imaging Spectroradiometer) vegetation data products MOD15A2 ($0.1^{\circ}*0.1^{\circ}$) |

[1] The *dw* in the unit means dry weight of biomass.

**Table 2**. Comparison between modelled and measured mean afternoon (12:00–18:00 UTC) mixing ratios of surface ozone at 537 rural AirBase stations. NE represents Northern Europe, CE Central Europe, and SE Southern Europe. MB – mean bias, ME – mean error, RMSE – root mean square error, MFB – mean fractional bias, MFE – mean fractional error.

| Season | Region | MB (ppb) | | ME (ppb) | | RMSE (ppb) | | MFB | | MFE | |
|--------|--------|------|-------|------|-------|------|-------|------|-------|------|-------|
| | | PSI | MEGAN | PSI | MEGAN | PSI | MEGAN | PSI | MEGAN | PSI | MEGAN |
| Winter | NE | -2.09 | -2.34 | 4.85 | 4.88 | 6.10 | 6.13 | -0.06 | -0.07 | 0.17 | 0.17 |
| | CE | 3.11 | 2.75 | 7.05 | 6.94 | 9.56 | 9.46 | 0.15 | 0.13 | 0.31 | 0.31 |
| | SE | 3.97 | 4.16 | 7.08 | 7.21 | 9.25 | 9.40 | 0.14 | 0.14 | 0.22 | 0.23 |
| | Total | 2.93 | 2.74 | 6.88 | 6.85 | 9.25 | 9.22 | 0.13 | 0.12 | 0.28 | 0.27 |
| Summer | NE | 4.76 | 5.27 | 6.74 | 6.96 | 8.62 | 8.90 | 0.15 | 0.16 | 0.20 | 0.21 |
| | CE | 1.97 | 2.70 | 6.55 | 6.34 | 8.53 | 8.33 | 0.08 | 0.09 | 0.18 | 0.17 |
| | SE | 0.68 | 2.20 | 6.82 | 6.80 | 9.03 | 9.03 | 0.04 | 0.06 | 0.16 | 0.15 |
| | Total | 1.82 | 2.76 | 6.64 | 6.52 | 8.68 | 8.57 | 0.07 | 0.09 | 0.17 | 0.17 |

**Table 3**. Statistical analysis of aerosols for different ACSM/AMS stations. MB - mean bias, ME – mean error, RMSE - root mean square error, MFB - mean fractional bias, MFE - mean fractional error.

| Species | Stations | Type | Time span (days) | MB ($\mu g\ m^{-3}$) | | ME ($\mu g\ m^{-3}$) | | RMSE ($\mu g\ m^{-3}$) | | MFB | | MFE | |
|---|---|---|---|---|---|---|---|---|---|---|---|---|---|
| | | | | PSI | MEGAN | PSI | MEGAN | PSI | MEGAN | PSI | MEGAN | PSI | MEGAN |
| OA | Zurich | Urban | Feb – Dec (324) | -1.41 | -4.31 | 3.56 | 4.51 | 4.88 | 5.82 | -0.28 | -0.90 | 0.63 | 0.95 |
| | Bologna | Urban | Nov– Dec ( 21) | -12.11 | -13.17 | 12.40 | 13.28 | 15.68 | 16.36 | -0.89 | -1.02 | 0.93 | 1.04 |
| | Marseille | Urban | Feb – Mar ( 24) | -6.13 | -6.97 | 6.18 | 6.98 | 8.20 | 8.95 | -1.05 | -1.19 | 1.08 | 1.22 |
| | Paris SIRTA | Suburban | Oct – Dec ( 92) | -6.24 | -6.50 | 6.29 | 6.53 | 9.51 | 9.68 | -1.08 | -1.35 | 1.12 | 1.36 |
| | Mace Head | Rural/Remote | Jan – Oct (328) | -0.09 | -0.53 | 0.44 | 0.53 | 1.04 | 1.25 | -0.80 | -1.64 | 1.15 | 1.66 |
| | San Pietro Capofiume | Rural/Remote | Nov– Dec ( 18) | -2.75 | -4.30 | 5.29 | 5.85 | 6.89 | 7.77 | -0.15 | -0.38 | 0.58 | 0.67 |
| | Montsec | Rural/Remote | Jul – Dec (171) | -1.93 | -2.45 | 2.00 | 2.46 | 2.69 | 3.20 | -1.01 | -1.42 | 1.04 | 1.42 |
| | Finokalia | Rural/Remote | Sep – Oct ( 30) | -1.23 | -2.48 | 1.56 | 2.48 | 2.20 | 3.10 | -0.55 | -1.20 | 0.71 | 1.21 |
| | SMEAR II Hyytiälä | Rural/Remote | Mar– Oct ( 30) | -0.08 | -0.46 | 0.49 | 0.54 | 0.87 | 0.95 | -0.13 | -0.64 | 0.60 | 0.78 |
| PSO$_4$ | Zurich | Urban | Feb – Dec (324) | 0.32 | 0.30 | 1.47 | 1.47 | 3.42 | 3.42 | 0.03 | 0.01 | 0.58 | 0.58 |
| | Bologna | Urban | Nov– Dec ( 21) | -0.10 | -0.06 | 1.92 | 1.94 | 2.53 | 2.54 | 0.04 | 0.05 | 0.60 | 0.60 |
| | Marseille | Urban | Feb – Mar ( 24) | 0.87 | 0.87 | 1.19 | 1.19 | 1.55 | 1.55 | 0.46 | 0.75 | 0.87 | 0.87 |
| | Paris SIRTA | Suburban | Oct – Dec ( 92) | 1.47 | 1.47 | 1.63 | 1.63 | 2.69 | 2.70 | 0.75 | 0.46 | 0.61 | 0.61 |
| | Mace Head | Rural/Remote | Jan – Oct (328) | 0.75 | 0.75 | 0.90 | 0.90 | 1.50 | 1.50 | 0.66 | 0.66 | 0.85 | 0.85 |
| | San Pietro Capofiume | Rural/Remote | Nov– Dec ( 18) | 2.21 | 2.26 | 2.27 | 2.32 | 2.95 | 3.00 | 0.91 | 0.92 | 0.94 | 0.95 |
| | Montsec | Rural/Remote | Jul – Dec (171) | 0.01 | -0.01 | 0.80 | 0.81 | 1.19 | 1.20 | 0.36 | 0.34 | 0.70 | 0.70 |
| | Finokalia | Rural/Remote | Sep – Oct ( 30) | -1.96 | -1.89 | 2.45 | 2.40 | 3.49 | 3.45 | -0.30 | -0.28 | 0.65 | 0.63 |
| | SMEAR II Hyytiälä | Rural/Remote | Mar– Oct ( 30) | 1.19 | 1.17 | 1.21 | 1.19 | 1.70 | 1.67 | 1.02 | 1.01 | 1.05 | 1.05 |
| PNO$_3$ | Zurich | Urban | Feb – Dec (324) | 1.88 | 1.87 | 2.93 | 2.95 | 4.35 | 4.49 | 0.52 | 0.49 | 0.96 | 0.95 |
| | Bologna | Urban | Nov– Dec ( 21) | 1.86 | 1.90 | 8.83 | 8.85 | 10.74 | 10.75 | 0.08 | 0.08 | 0.72 | 0.72 |
| | Marseille | Urban | Feb – Mar ( 24) | 1.93 | 2.01 | 3.12 | 3.20 | 4.03 | 4.13 | 0.54 | 0.76 | 1.00 | 1.00 |
| | Paris SIRTA | Suburban | Oct – Dec ( 92) | 2.14 | 2.13 | 2.91 | 2.91 | 4.32 | 4.32 | 0.76 | 0.54 | 0.91 | 0.92 |
| | Mace Head | Rural/Remote | Jan – Oct (328) | 1.07 | 1.21 | 1.07 | 1.21 | 2.81 | 3.18 | 1.46 | 1.48 | 1.48 | 1.50 |
| | San Pietro Capofiume | Rural/Remote | Nov– Dec ( 18) | 7.93 | 7.85 | 10.54 | 10.45 | 13.12 | 13.06 | 0.76 | 0.75 | 1.05 | 1.05 |
| | Montsec | Rural/Remote | Jul – Dec (171) | -0.10 | -0.10 | 0.50 | 0.50 | 0.85 | 0.85 | 0.08 | 0.08 | 1.05 | 1.05 |
| | Finokalia | Rural/Remote | Sep – Oct ( 30) | 0.31 | 0.34 | 0.34 | 0.37 | 0.68 | 0.80 | 0.61 | 0.61 | 0.96 | 0.96 |
| | SMEAR II Hyytiälä | Rural/Remote | Mar– Oct ( 30) | 1.89 | 2.10 | 1.89 | 2.10 | 2.66 | 2.90 | 1.71 | 1.74 | 1.71 | 1.74 |
| PNH$_4$ | Zurich | Urban | Feb – Dec (324) | 1.05 | 1.04 | 1.27 | 1.27 | 2.24 | 2.28 | 0.61 | 0.56 | 0.82 | 0.80 |
| | Bologna | Urban | Nov– Dec ( 21) | 0.53 | 0.56 | 2.81 | 2.83 | 3.45 | 3.46 | 0.07 | 0.08 | 0.64 | 0.64 |
| | Marseille | Urban | Feb – Mar ( 24) | 0.39 | 0.41 | 1.01 | 1.02 | 1.34 | 1.36 | 0.24 | 0.44 | 0.78 | 0.78 |
| | Paris SIRTA | Suburban | Oct – Dec ( 92) | 0.90 | 0.89 | 1.17 | 1.17 | 1.82 | 1.82 | 0.44 | 0.24 | 0.59 | 0.60 |
| | Mace Head | Rural/Remote | Jan – Oct (328) | 0.44 | 0.48 | 0.49 | 0.53 | 1.20 | 1.30 | 0.33 | 0.35 | 1.14 | 1.15 |
| | San Pietro Capofiume | Ruran/Remote | Nov– Dec ( 18) | 2.74 | 2.73 | 3.29 | 3.27 | 4.21 | 4.20 | 0.76 | 0.76 | 0.92 | 0.91 |

| Species | Stations | Type | Time span (days) | MB ($\mu$g m$^{-3}$) | | ME ($\mu$g m$^{-3}$) | | RMSE ($\mu$g m$^{-3}$) | | MFB | | MFE | |
|---|---|---|---|---|---|---|---|---|---|---|---|---|---|
| | | | | PSI | MEGAN | PSI | MEGAN | PSI | MEGAN | PSI | MEGAN | PSI | MEGAN |
| | Montsec | Ruran/Remote | Jul – Dec (171) | -0.27 | -0.28 | 0.38 | 0.39 | 0.60 | 0.61 | -0.32 | -0.25 | 0.63 | 0.72 |
| | Finokalia | Ruran/Remote | Sep – Oct ( 30) | -0.47 | -0.45 | 0.66 | 0.65 | 0.96 | 0.94 | -0.30 | -0.28 | 0.60 | 0.59 |
| | SMEAR II Hyytiälä | Rural/Remote | Mar– Oct ( 30) | 0.84 | 0.88 | 0.85 | 0.89 | 1.18 | 1.23 | 1.04 | 1.07 | 1.25 | 1.27 |

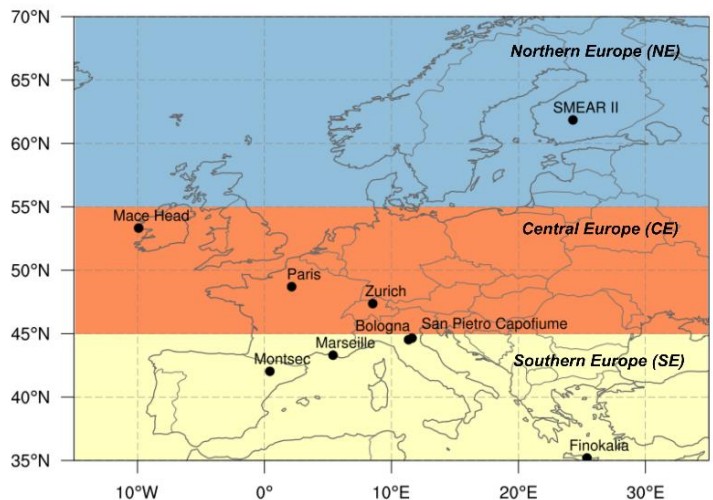

**Figure 1.** Model domain and location of ACSM/AMS measurement stations.

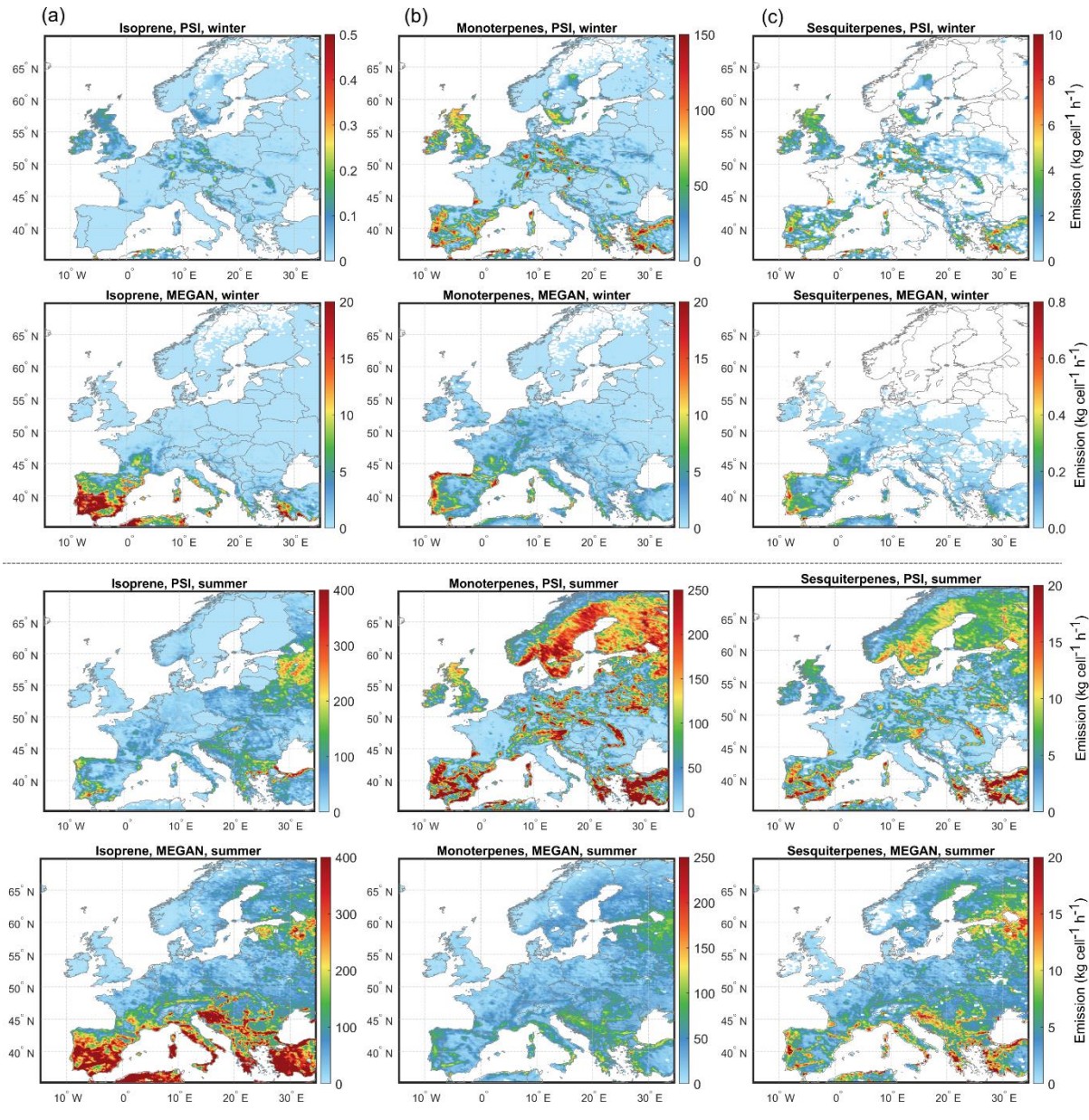

**Figure 2.** Average hourly emissions of isoprene **(a)**, monoterpenes **(b)** and sesquiterpenes **(c)** estimated by PSI model and MEGAN v2.1. Upper and lower panels represent winter and summer cases, respectively.

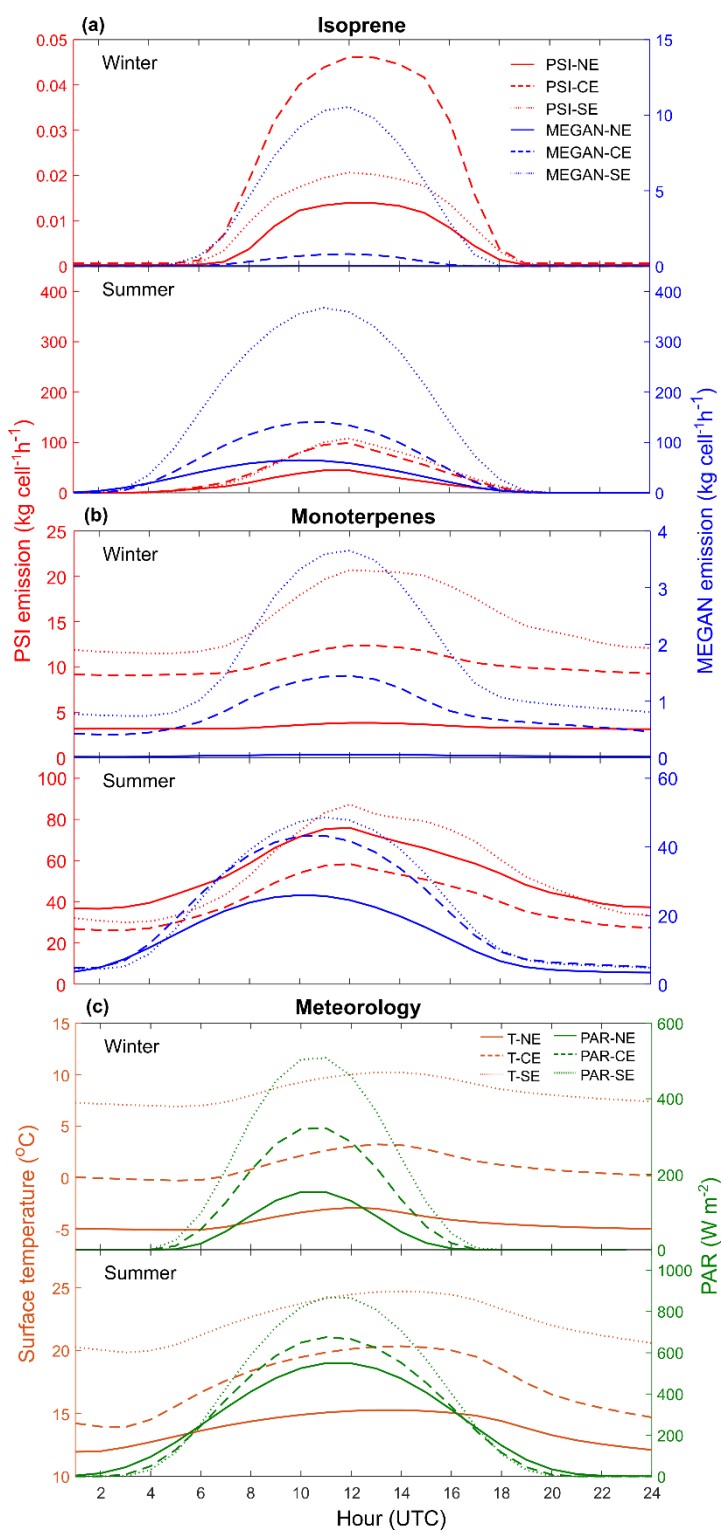

**Figure 3.** Diurnal variations of average grid-scale isoprene (a), monoterpene emissions (b) in the model domain estimated by PSI model (y-axis left) and MEGAN v2.1 (y-axis right), and meteorological conditions (c). NE represents Northern Europe, CE Central Europe, and SE Southern Europe.

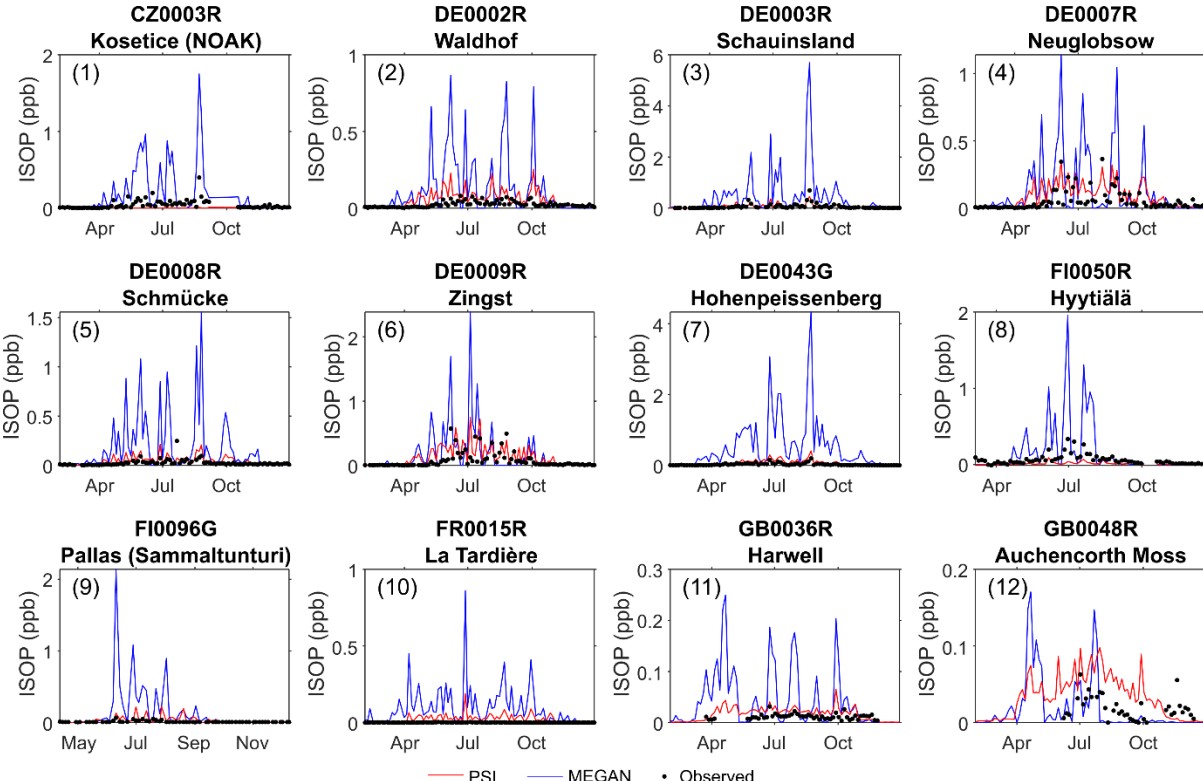

**Figure 4.** Comparison between modelled and measured isoprene concentrations in 2011. The measurement data were obtained from EBAS database (http://ebas.nilu.no/) operated by Norwegian Institute for Air Research (NILU). The time resolution of measurements varies with sites: at station 9 FI0096G every 72 hours, at stations 1−6 and 10 every 96 hours, while at station 7, 8, 11 and 12 every 3−12 hours but averaged to 96 hours for better visualization.

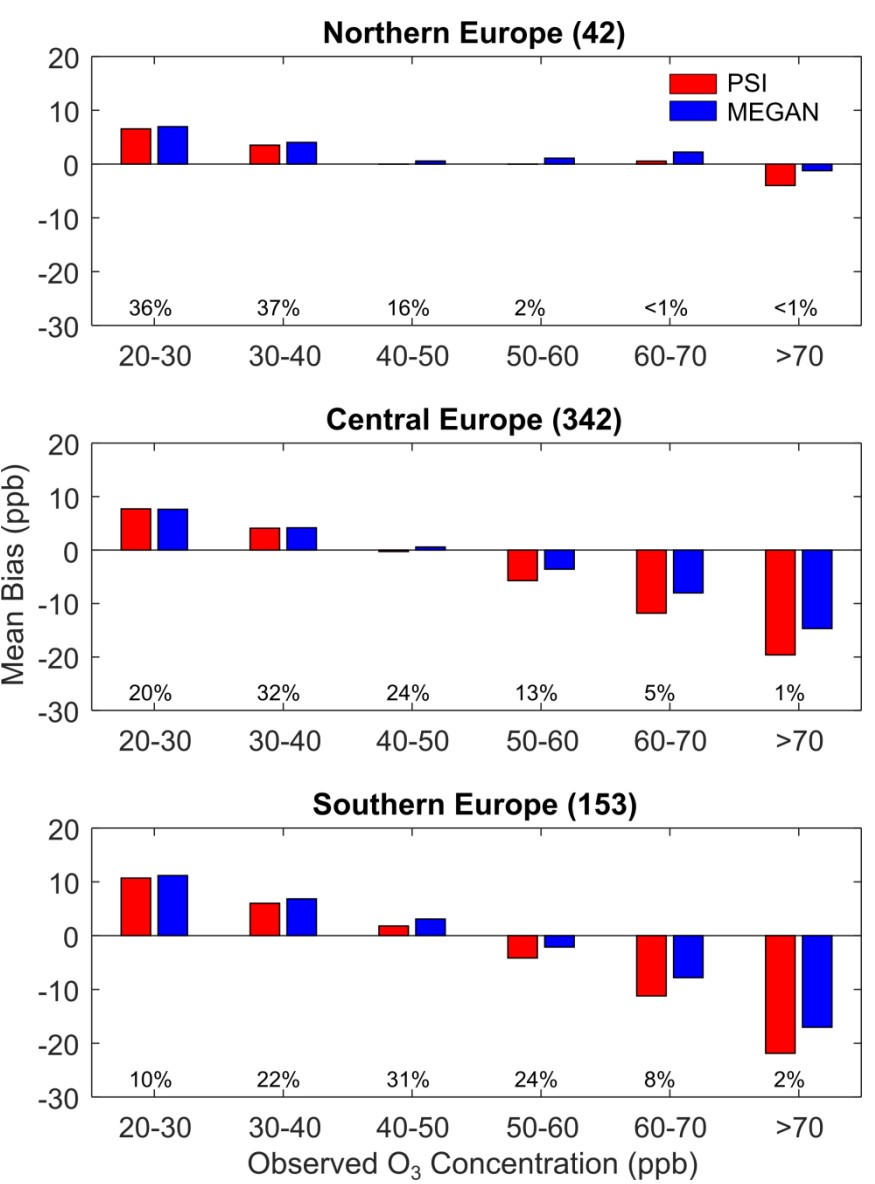

**Figure 5.** Mean bias of surface $O_3$ mixing ratios in the afternoon (12:00–18:00 UTC) for each bin of observed hourly average ozone in July 2011. The number of stations available for each region is reported in parentheses at the top of each panel. Percentage values below the bars show the relative fraction of data in each bin.

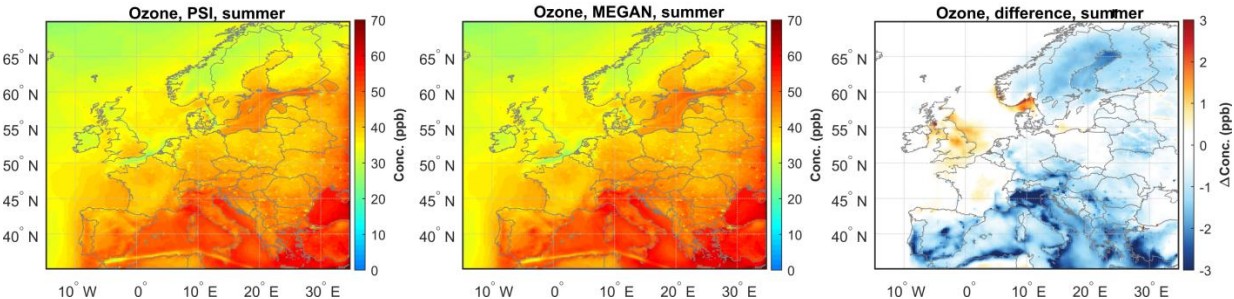

**Figure 6.** Modelled afternoon (12:00–18:00 UTC) mixing ratios of surface ozone in summer using PSI emissions (O$_3$-PSI, left), MEGAN emissions (O$_3$-MEGAN, middle) and the difference between O$_3$-PSI and O$_3$-MEGAN (right).

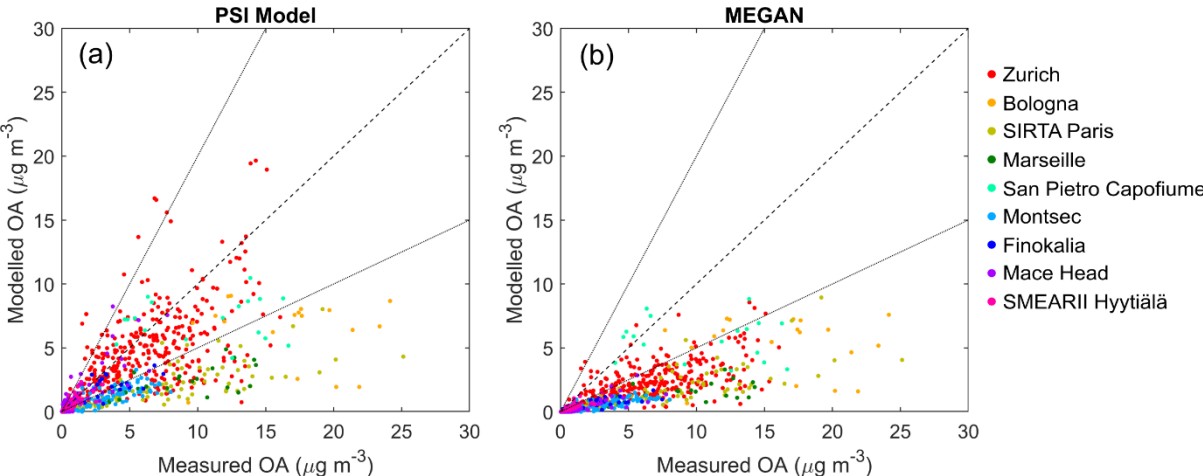

**Figure 7**. Modelled versus measured daily OA concentrations using BVOC emissions calculated by PSI model (OA-PSI) **(a)** and MEGAN (OA-MEGAN) **(b)** at 9 ACSM/ AMS stations. The dashed line represents 1:1 line, dotted lines represent 2:1 and 1:2 lines.

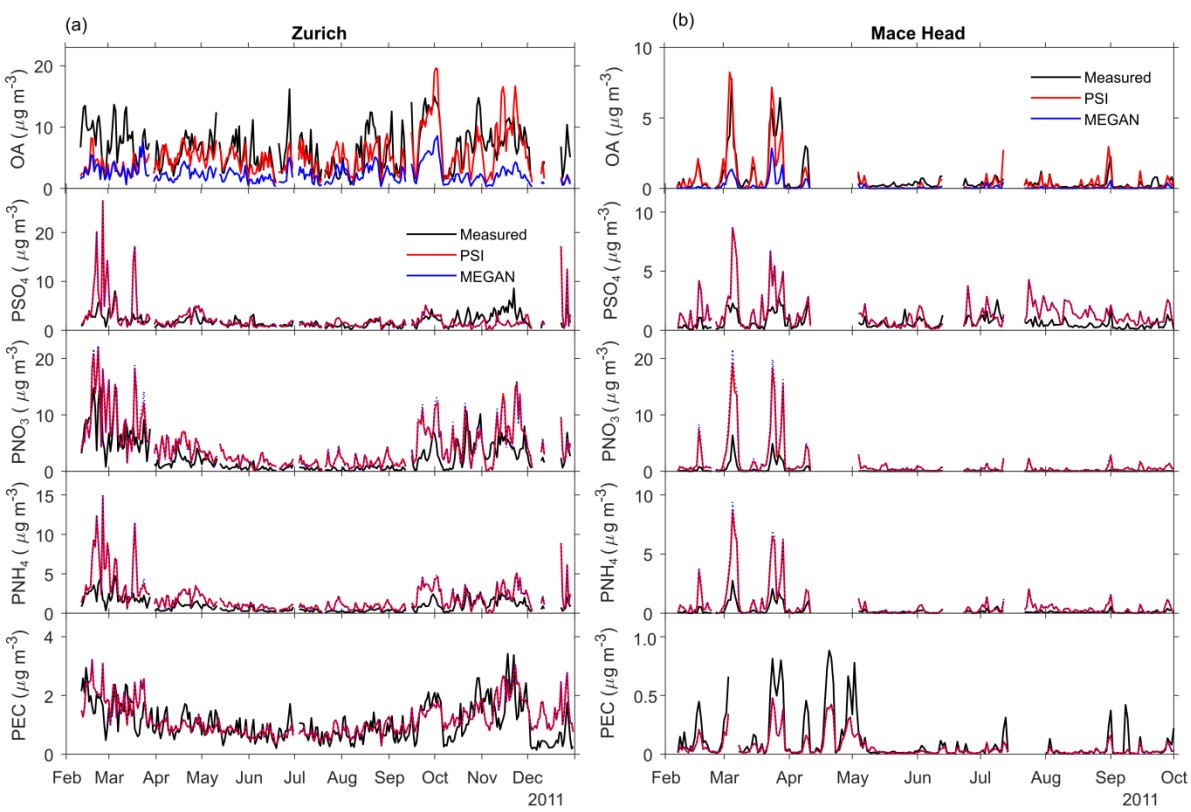

**Figure 8**. Temporal variation of the modelled (with both PSI and MEGAN emissions) and measured concentrations of organic and inorganic aerosols at Zurich **(a)** and Mace Head **(b)** in 2011.

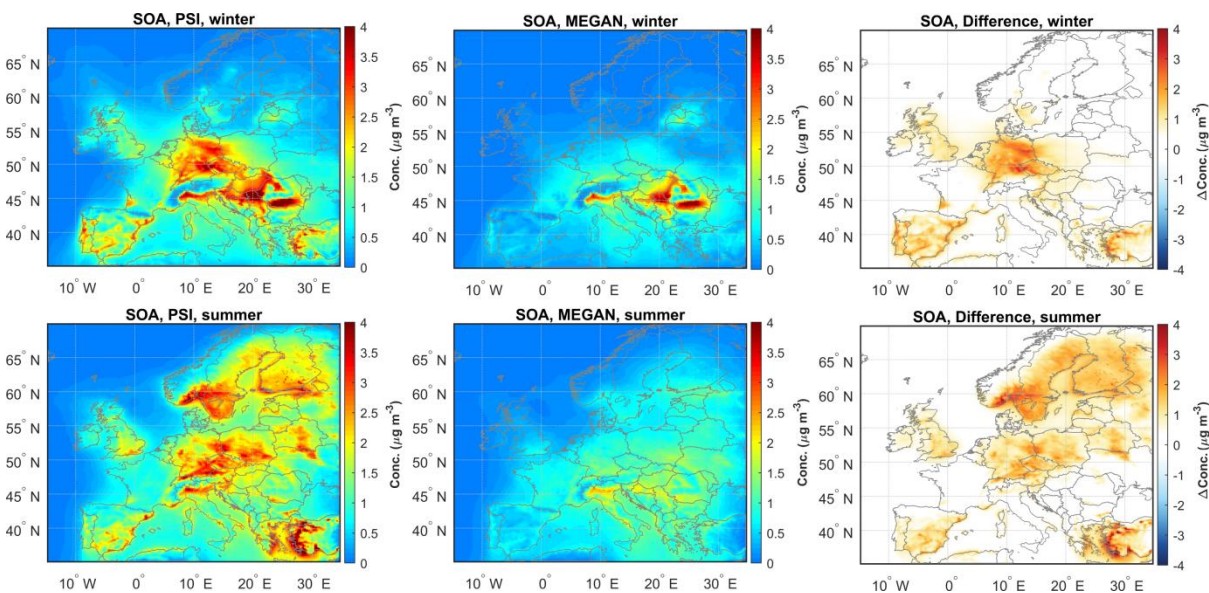

**Figure 9.** Modelled SOA concentrations using PSI emissions (SOA-PSI) (left), MEGAN emissions (SOA-MEGAN) (middle) and the difference between SOA-PSI and SOA-MEGAN (right).

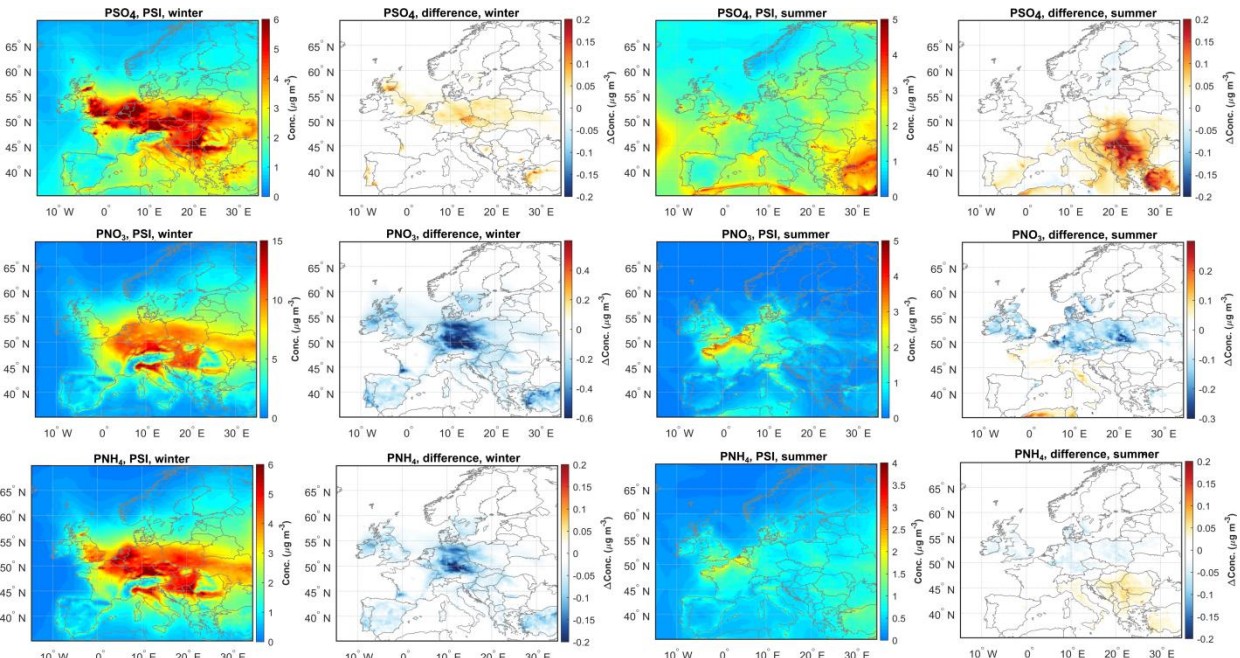

**Figure 10.** Modelled secondary inorganic aerosol (SIA) concentrations using PSI emissions and the difference between PSI and MEGAN.