# Peer review of "Effects of two different biogenic emission models on modelled ozone and aerosol concentrations in Europe"

_Atmospheric Chemistry and Physics, 2018_

## Referee Comment (RC1) · Anonymous Referee #1 · 19 Nov 2018

Jiang et al. describe a modeling study comparing two biogenic VOC emission models, MEGAN and PSI, and their effects on modeled ozone and aerosols in Europe using a chemical transport model CAMx. The two BVOC models mainly differ in vegetation classification and reference basal emission rates. PSI predicts much lower isoprene emission but 3 times of monoterpene emissions higher than MEGAN. Such emission differences result in relatively small differences in ozone (<10%) but very large differences in SOA.

The manuscript is well structured and generally clearly stated. The study focuses on the impact of different BVOC inputs which is one of the fundamentals of atmospheric

chemistry. I recommend publication of this manuscript in ACP after minor revisions.

My major concern is that the two BVOC models predict very different patterns and magnitude of isoprene and monoterpene emissions, but readers have no idea about how good they are compared to real observations. I suggest to add a section comparing the PSI and MEGAN results with at least some in-situ measurement of isoprene and monoterpene emissions. Validation of only ozone and SOA are not enough to fully understand the strengths and weaknesses of the two models, as many other factors may contribute to the formation of ozone and SOA and they may compensate each other.

Other comments: P6 L1: Are those factors including soil moisture and $CO_2$ dependence "turned on" in your simulations?

Figure 2: Font of legends should be consistent.

Figure 3: How to interpret different MT peak time in MEGAN and PSI (in summer, bottom panel), even though they adopt a similar T-dependent function and use the same meteorology input?

P8 L23-25: Better to give some rough numbers of these model-observation comparisons from these references.

P9 L14: The statement "the spatial difference in simulated O3 and isoprene emissions" is not clear. What variables are used here to calculate the correlation?

P9 L26-28: Can you provide more information on NOx and ozone background concentration? Is the whole European domain within the NOx-sensitive regime?

P10 L9-12: Can you add two lines/shades to represent primary and biomass burning OA in Figure 7? It would be more straightforward to see the contributions of biogenic versus other sources.

P11 L27: "vertical distribution of elevated emissions" should be "vertical ventilation"?

---

## Referee Comment (RC2) · Anonymous Referee #2 · 19 Nov 2018

**GENERAL**

Understanding sources of uncertainties in ozone and SOA simulations are important steps for improving air pollution modelings. This study compares two different BVOC schemes and the consequent impacts on ozone and SOA in Europe. The differences between PSI and MEGAN schemes are discussed. The authors found that PSI scheme predicts more monoterpenes while the MEGAN scheme predicts more isoprene. As a result, the CTM based on PSI yields more SOA than the results based on MEGAN scheme. The topic of this study well fits the scope of ACP journal, however, some essential limits may largely weaken the scientific merits of the work.
First, the differences in BVOC are most likely attributed to those in land cover instead of schemes. In general, PSI uses an earlier version of MEGAN parameterization for isoprene and a current MEGAN parameterization for monoterpene. They should have similar responses to environmental factors such as light and temperature. The main reason why PSI and MEGAN schemes show such a large difference in BVOC emissions is that they use different land cover. The authors clarified that PSI is based on tree species while MEGAN is based on PFTs. What if the MEGAN scheme uses the same land cover as PSI, but with PFTs aggregated from tree species? The land cover should be uniform before the comparison.

Second, no BVOC observations are used to constrain simulations. Though the authors use the measurements of ozone and OA to validate model results, these are not the direct observations of BVOC. One can simulate right air pollution with wrong reasons (e.g., poor model performance, incorrect meteorology and so on). The only way to check the validity of BVOC schemes is to compare simulations with direct measurements of isoprene and/or monoterpene, which I believe there are many over Europe. Without BVOC constraints, the current study is more like a sensitivity test of ozone and SOA in CTM to any perturbations in BVOC emissions.

**SPECIFIC**

Page 1, line 32: "improving substantially the model performance", How do you know it improves the model for correct reason?

Page 2, Line 38: "highest over all the model inputs" What kind of inputs? Specify.

Page 4, Line 3: "Initial and boundary conditions were ...." What kind of IC and BC? Specify.

Page 5, Line 31: "The value of 0.1 is used for MT in MEGAN2.1, while the values are between 0.065 to  $0.077 \dots$  in PSI model" Why the PSI model uses different parameters while it uses the same scheme as MEGAN?
Page 6, Line 3: "canopy model" What's the impacts of different canopy models on the simulated light availability for PSI and MEGAN models?

Page 6, Line 30: "The variation of biomass density in MEGAN was simulated by the satellite data" How the satellite data simulates biomass density?

Page 7, Line 27: "To demonstrate the seasonal differences". The word "demonstrate" is not appropriate, better to use "evaluate" or "quantify".

Page 8, Line 7: "observed" This is not observation. Better to use "found"

Page 8, Line 14: "In winter, highest isoprene emissions occurred in Central Europe for PSI model" Why there are isoprene emissions in winter when leaf biomass is set to zero.

Page 9, Line 13: "observed", again not observation. Better to use "calculated"

Page 9, Line 28: "background", where is the background ozone from?

Page 10, Line 11: "and" should be "but"

Page 11, Section 3.2.3: Not sure whether this section is necessary as BVOC has minor impacts on SIA

Figures 3 and 4: Why in CE, isoprene is much higher for PSI but ozone is still lower than MEGAN?

**ACPD**

---

## Referee Comment (RC3) · Anonymous Referee #3 · 26 Nov 2018

**Review Summary**

Jiang et al. simulated ozone and aerosol concentrations in Europe using two different biogenic emission models (PSI and MEGAN) to probe uncertainties in regional air quality models. They compared model results with ozone observations from the European air quality database, AirBase, and aerosol observations from eight different measurement locations with an Aerodyne AMS or ACSM. Results were generally consistent with previously published papers demonstrating that MEGAN tends to over-estimate isoprene and under-estimate monoterpene emissions. They also found that the simulated ozone mixing ratios between the model runs varied less than the isoprene emissions.

sions. This is also consistent with previous studies showing much of Europe's ozone production is NOx-limited rather than VOC-limited. Finally, their model comparison suggests higher monoterpene emissions lead to better comparison between simulated and observed organic aerosol. The authors acknowledge this could be due to compensating factors (e.g. they could be "right for the wrong reasons"). Overall, the scientific approach is reasonable and the scientific questions are appropriate for the scope of the journal. However, it is unclear what information this paper is adding to the scientific community that has not already been published in previous papers. There are also a number of gaps in the methods section that lack clarity. I recommend publication after the manuscript is revised to address the following comments.

General Comments The authors should better clarify how this particular paper is filling in gaps that have not already been addressed in previous publications. All results sections generally state the results are consistent with work that has already been published, and so it is very unclear what the conclusions from this paper are adding to the growing body of scientific knowledge. The manuscript could better highlight how this work is filling in unique gaps in understanding.

Specific Comments METHODS: SECTION 2.2.1 EMISSION RATES Authors state they estimate reference emission rates of isoprene and monoterpenes based on Lamb et al., 1993 but then go on to say Norway spruce isoprene emissions were estimated to be 10% of alpha-pinene. It is unclear why Norway spruce was handled differently and why it is singled out to be described separately from the other plant species. Please clarify.

The PSI model emission rates are species-specific except for "pasture" and "crop" (Table 1). How much variability would you expect between different types of "pasture vegetation" and "crop vegetation" based on the literature? What proportion of the total area covered in the model is characterized as "pasture" and "crop"? Is it a significant portion of the land that could drastically impact results or is it minor?
Sesquiterpene emissions: authors state that sesquiterpene emissions were assumed to be 5% (by weight) of monoterpene emissions based on field measurements from various studies, but then cite a single paper that is actually a modelling paper and not a review or synthesis of measurements. Please cite the original literature from which this "5% (by weight)" reference is derived.

METHODS: SECTION 2.2.2, RESPONSE FUNCTIONS If sesquiterpenes are being treated as pooled emissions as stated in the previous section, then they will be treated similar to monoterpenes. However, the authors do not discuss what  $\beta$  value they used for sesquiterpenes. In Guenther et al., (2012) the empirically-derived temperature correction coefficient,  $\beta$ , for sesquiterpenes was 0.17. Was that the value used in this study as well? Please clarify.

Again, the authors single out Norway spruce emissions being handled a bit differently than other plant species. In this case, the Norway spruce monoterpene emissions have some light-dependent fraction estimated based on a study in 1993. Why is Norway spruce being singled out for more detailed emission estimation? Is it the dominant species in the modeling domain? This should be clarified. Also, more details should be included about how the light-dependent emissions were estimated instead of just referring to the 1993 paper with no summary of what information was taken from that paper and used in this study. Finally, does this section then imply that all other monoterpene emissions were light-independent? Can this be stated more clearly and justified? If all monoterpene emissions are being treated as light-independent (except for some unstated fraction of Norway spruce monoterpene emissions), then this should be justified because it is well known that a substantial fraction of monoterpene emissions are light-dependent fraction of monoterpene emissions are light-dependent fraction of monoterpene emissions are light-dependent fraction of monoterpene emissions are light-dependent; for example, in MEGANv2.1 the light-dependent fraction of monoterpene emissions ranges from 40-80%! (see Guenther et al., 2012, Table 4).

**METHODS SECTION 2.2.3 INPUTS OF DRIVING VARIABLES**

Unclear how GlobCover 2006 data is being used to derive species-level distributions.
How did the authors go from fractions of "needleleaf, broadleaf and mixed forests" to plant species distribution using the profiles from Simpson et al., 1999? There is missing information here that links the two.

How much of variation between PSI and MEGAN emissions was driven by differences in normalizing emissions to leaf surface area (MEGAN) versus leaf biomass (PSI)? Are there potential biases that could vary between plant types for comparing total canopy-scale flux that arise from how surface area versus biomass are scaled up? How did the authors ensure they were making meaningful comparisons between the models with the emissions normalized differently? Figure 2 was clear: authors graphed the emission rate per model grid cell. It was less clear how this comparison was done in Figure 3 where the graph simply shows the emission rate. Was this also per model grid? Per entire modeling domain? This should be stated more clearly.

Figure 2: right axis label is cut off on third row.

Figure 4: Figure caption should be re-worded. Currently states, "Mean bias of surface O3 mixing ratios in the afternoon 12:00-18:00 UTC) for each bin of observed ones in July 2011." I suggest revising to more clearly describe what is meant by "observed ones".

RESULTS O3: results are consistent with previous literature demonstrating that O3 production in most of Europe is in a NOx-limited regime as opposed to a VOC-limited regime and thus the isoprene differences between the two models do not translate into large differences in ozone. Not a novel result. Can the authors comment on how this study is different from previous ones that have published the same result?

OA comparison: the study does not have a single AMS/ACSM location in the Northern Europe region. Surely there are measurements at Hyytiälä, or some other boreal forest site in northern Europe. Can you justify why no measurement sites were included for northern Europe?

**ACPD**
SECTION 3.2.3: INORGANIC AEROSOLS Authors do not set up a rationale in the introduction for investigating the impact of changing biogenic emissions on inorganic aerosols. Why would differences in biogenic emissions substantially alter inorganic aerosol? Without this rationale in the introduction, this section does not fit with the rest of the paper.

DISCUSSION Authors end the paper by saying, "In future studies, BVOC emission models with more regional specific adaptation in vegetation types and emission factors are urgently needed to reduce the uncertainties in BVOC emission estimates in order to improve air quality modelling." Why is this the recommendation rather than simply improving the emission factors for the plant functional types in MEGAN? It is not reasonable to model the emissions from every single plant species on the planet. I don't agree that the results from this study emphasize the need for more plant specificity because even this paper only used a sub-set of 10 specific plant species (with an additional two general classes for "pasture" and "crop"). It seems to me that the major finding from this paper, consistent with published papers before it, is that the MEGAN emission factors could be updated and improved.

---

## Author Comment (AC1) · 28 Dec 2018

**Responses to the comments of anonymous referee #1**

We thank the referee for the valuable comments which have greatly helped us to improve the manuscript. Please find below our responses (in black) after the referee comments (in blue). The changes in the revised manuscript are written in *italic*.

Jiang et al. describe a modeling study comparing two biogenic VOC emission models, MEGAN and PSI, and their effects on modeled ozone and aerosols in Europe using a chemical transport model CAMx. The two BVOC models mainly differ in vegetation classification and reference basal emission rates. PSI predicts much lower isoprene emission but 3 times of monoterpene emissions higher than MEGAN. Such emission differences result in relatively small differences in ozone (<10%) but very large differences in SOA. The manuscript is well structured and generally clearly stated. The study focuses on the impact of different BVOC inputs which is one of the fundamentals of atmospheric chemistry. I recommend publication of this manuscript in ACP after minor revisions.

My major concern is that the two BVOC models predict very different patterns and magnitude of isoprene and monoterpene emissions, but readers have no idea about how good they are compared to real observations. I suggest to add a section comparing the PSI and MEGAN results with at least some in-situ measurement of isoprene and monoterpene emissions. Validation of only ozone and SOA are not enough to fully understand the strengths and weaknesses of the two models, as many other factors may contribute to the formation of ozone and SOA and they may compensate each other.

For the evaluation of the model we prefer to do this on more stable species like the secondary products such as ozone and SOA. The BVOC concentrations are strongly influenced by local mixing processes and chemistry due to the high reactivity of these molecules. The model output is unlikely representative for species with strong spatial gradients. Also, such measurements are very sparse. In spite of these considerations we agree with the referee that after all it is important to give the reader some idea about the BVOC model performance. We added a figure about the comparison of modelled and measured isoprene in the revised manuscript (see new Fig. 4). The only measurements of some monoterpene species during the simulated period were in Finland. We compared them with our modelled total monoterpene concentrations. We also compared our results with some measurements reported in the literature for other years. We inserted the following text in Section 3.1 (P10, L3-L24) of the revised manuscript:

*„BVOC measurements are rare and the concentrations are associated with very high spatial gradients (especially vertical) due to high reactivity and local mixing processes that are unlikely captured by the model in the respective grid cell. Nevertheless but with these caveats in mind we compared a few measurements available for isoprene with our model results to get an idea about the range of differences. Compared to monoterpenes, there were more isoprene measurements at various European sites in 2011 (see Fig. 4). Clearly, the MEGAN-isoprene data are much higher than the measurements at all 12 sites while the PSI- isoprene results are closer to the measurements.*

*Unlike the single compound of isoprene, monoterpenes consist of several species and therefore it is even more difficult to perform comparisons with measurements, which are rare and have large uncertainties. Only a limited number of MT measurements were reported in Europe (only in Finland) during our simulation period (Hakola et al., 2012; Hellen et al., 2012). Hakola et al. (2012) reported average MT concentrations of about 508 ppt (with a range between about 150 and 800 ppt) in August 2011 at the SMEAR II station at Hyytiälä. MEGAN-MT for the same period was 117 ppt while PSI-MT was around 2 ppb (for the same site, Rinne et al. (2005) reported MT concentrations of between 200-500 ppt during daytime and more than 1 ppb at nighttime in summer 2004). On the other hand, the measured MT concentrations at a nearby urban background station SMEARIII in Helsinki were lower, with around 117*

*ppt in summer (Hellen et al., 2012). Both models predicted higher concentrations for that site (MEGAN-MT 303 ppt, PSI-MT 1 ppb). In order to get an idea about the model performance in other regions, we compared our results also with MT concentrations measured at Hohenpeissenberg (southern Germany) in June 2006 (Oderbolz et al., 2013). Both model results (PSI-MT: 75 ppt, MEGAN-MT: 130 ppt) in that region were similar to measurements (~100 ppt). Although this comparison of measurements and model results for different years under different meteorological conditions has a very high uncertainty, it might help to understand the range of differences between the model results and the measurements. In general, all these comparisons suggest that MT concentrations might be underestimated using MEGAN emissions while PSI emissions might be too high over Scandinavia. On the other hand, both models seem to predict MT emissions relatively well in central Europe.*"

Other comments:

P6 L1: Are those factors including soil moisture and $CO_2$ dependence "turned on" in your simulations?

Since we are using the offline version of MEGAN v2.1, the soil moisture and $CO_2$ dependence corrections were not included (Emmerson et al., 2016). We used the default parameterization where these factors were set to 1. The $CO_2$ inhibition effect might be significant in regions with high $CO_2$ and isoprene emissions. Studies using global coupled land-atmosphere models reported that accounting for $CO_2$ inhibition has little impact on predictions of present-day global isoprene emissions but might have larger effects on future emissions (Heald et al., 2009, Tai et al., 2013). We rephrased the sentence in the revised manuscript (P6 L23-L25) as follows:

"*In addition to the light and temperature response, MEGAN v2.1 covers also some other factors such as leaf age and leaf area index (Guenther et al., 2012). Since the correction of soil moisture and $CO_2$ dependence are not included in the offline version of MEGAN (Emmerson et al., 2016), we used the default parameterization where the correction factors were set to 1.*"

Figure 2: Font of legends should be consistent.

Corrected

Figure 3: How to interpret different MT peak time in MEGAN and PSI (in summer, bottom panel), even though they adopt a similar T-dependent function and use the same meteorology input?

In addition to the T-dependent pool emissions, both the PSI model and MEGAN include species having both light and temperature dependent synthesis emissions. Different fractions of the light-dependent MT emissions of the two models could lead to different MT peak times. We added the diurnal variation of T and PAR to Figure 3 in the revised manuscript to show the different T/PAR dependence of MT emissions of two models. We updated Section 2.2.2 (P6 L19-L22) to clarify the influence of light-dependent response to MT emissions as follows:

"*The light-dependent synthesis emissions of MTs were considered in MEGAN v2.1 as described in Guenther et al. (2012). Depending on different MT species, the light-dependent fraction of MT emissions ranges between 0.2 to 0.8 for MEGAN. In the PSI model, the light-dependent emissions from Norway spruce are calculated for each monoterpene species as a function of PAR based on the data of Schürmann (1993).*"

We also added an explanation about the different MT peak times in Section 3.1 (P9 L27-L31) as follows:

"*Comparison of monoterpenes emissions (Fig. 3b) with temperature and photosynthetically active radiation (PAR) (Fig. 3c) indicates that monoterpene emissions by the PSI model are mostly temperature-dependent while the influence of light is stronger for the MEGAN–MT emissions. For instance, the highest PSI–MT emissions in summer occurred at the same time of the highest temperature (13:00–14:00 UTC), while the occurrence of highest MEGAN–MT is close to the PAR peak (10:00–12:00 UTC).*"

We revised the sentence (P10 L25-L28) as follows:

*"Studies comparing different models with each other, as well as with measurements suggest that MEGAN tends to overestimate isoprene emissions especially in Scandinavian countries and south-west Europe and to underestimate monoterpene emissions by more than a factor of 2 (Bash et al., 2016; Carlton and Baker, 2011; Emmerson et al., 2016; Poupkou et al., 2010; Silibello et al., 2017)."*

We rephrased this sentence (P11 L21-L23) to clarify it as follows:

*"The spatial distribution of the ozone difference, i.e. (PSI-$O_3$) – (MEGAN-$O_3$) (Fig. 6, right panel) is very similar to that of the difference in the isoprene emissions (Fig. S2a)"*

Several European studies reported that ozone formation in most regions is $NO_x$-sensitive in general except around the English Channel, Benelux and Po Valley regions, where $NO_x$ emissions are higher and the response to a change in the VOC emissions is relatively stronger (Beekman and Vautard, 2010; Aksoyoglu et al., 2012; Oikonomakis et al., 2018). We added some discussion in P11 L34 to P12 L11.

*"The main reason for the weak effect of the isoprene emissions on ozone is the stronger sensitivity of ozone formation in general to $NO_x$ emissions rather than VOC emissions in Europe. An additional reason might be the rather low ozone production compared to the background ozone where the latter is not affected by local European emissions (Oikonomakis et al., 2018; Sartelet et al., 2012). Several European studies reported that ozone formation in most regions is $NO_x$-sensitive except around the English Channel, Benelux and Po Valley regions, where $NO_x$ emissions are high (due to intensive anthropogenic NOx emissions from both land and shipping or geographical characteristics leading to high accumulation of pollutants) and the response to a change in the VOC emissions is relatively stronger (Aksoyoglu et al., 2012; Beekmann and Vautard, 2010; Oikonomakis et al., 2018). However, the sensitivity of ozone formation to its precursor emissions might change as a result of large $NO_x$ emission reductions in Europe since 1990 according to the Gothenburg Protocol. On the other hand, emissions from shipping activities are not regulated as strictly as land emissions and have been increasing continuously especially in the Mediterranean, affecting both ozone and particulate matter concentrations (Viana et al., 2014; Aksoyoglu et al., 2016)."*

We totally agree that showing the contribution of biogenic versus anthropogenic sources would be more straightforward. However, this is a topic of another manuscript (in prep.) in which we focus on the source apportionment of organic aerosols, therefore we prefer not to show such figures in this manuscript. In order to reply the referee's question however, we show below (Fig. 1) the anthropogenic and biogenic OA concentrations (stacked) modelled using the PSI and MEGAN emissions at two sites. These figures show that the contribution of biogenic emissions to OA is higher with the PSI emissions at both sites. The modelled fractions of biogenic and anthropogenic OA were found to be closer to the PMF analysis of the measured data at Zurich (Canonaco et al., 2013; Daellenbach et al., 2017) when the PSI emissions were used.

[Figure]

Fig. 1: Time series of anthropogenic and biogenic OA modelled by using PSI and MEGAN BVOC emissions at Zurich (left) and Mace Head (right).

P11 L27: "vertical distribution of elevated emissions" should be "vertical ventilation"?
What we mean is the injection of point-source emissions into the vertical layers of the model domain. We revised the sentence (P14 L14-L15) to make it clearer as follows:

*"The precursor gases $SO_2$ and $NO_x$ from anthropogenic point sources (continental, shipping) (Fig. S8) might be accumulated too much in the surface layer since all emissions were injected into the $1^{st}$ model layer, leading to too high SIA formation. "*

**Reference**

Aksoyoglu, S., Keller, J., Oderbolz, D. C., Barmpadimos, I., Prévôt, A. S. H., and Baltensperger, U.: Sensitivity of ozone and aerosols to precursor emissions in Europe, Int. J. Environ. Pollut., 50, 451-459, 10.1504/ijep.2012.051215, 2012.

Aksoyoglu, S., Baltensperger, U., and Prevot, A. S. H.: Contribution of ship emissions to the concentration and deposition of air pollutants in Europe, Atmos. Chem. Phy., 16, 1895-1906, 10.5194/acp-16-1895-2016, 2016.

Bash, J. O., Baker, K. R., and Beaver, M. R.: Evaluation of improved land use and canopy representation in BEIS v3.61 with biogenic VOC measurements in California, Geosci. Model Dev., 9, 2191-2207, 10.5194/gmd-9-2191-2016, 2016.

Beekmann, M., and Vautard, R.: A modelling study of photochemical regimes over Europe: robustness and variability, Atmos. Chem. Phy., 10, 10067-10084, 10.5194/acp-10-10067-2010, 2010.

Canonaco, F., Crippa, M., Slowik, J. G., Baltensperger, U., and Prévôt, A. S. H.: SoFi, an IGOR-based interface for the efficient use of the generalized multilinear engine (ME-2) for the source apportionment: ME-2 application to aerosol mass spectrometer data, Atmos. Meas. Tech., 6, 3649-3661, 10.5194/amt-6-3649-2013, 2013.

Carlton, A. G., and Baker, K. R.: Photochemical Modeling of the Ozark Isoprene Volcano: MEGAN, BEIS, and Their Impacts on Air Quality Predictions, Environ. Sci. Technol., 45, 4438-4445, 10.1021/es200050x, 2011.

Daellenbach, K. R., Stefenelli, G., Bozzetti, C., Vlachou, A., Fermo, P., Gonzalez, R., Piazzalunga, A., Colombi, C., Canonaco, F., Hueglin, C., Kasper-Giebl, A., Jaffrezo, J. L., Bianchi, F., Slowik, J. G., Baltensperger, U., El-Haddad, I., and Prévôt, A. S. H.: Long-term chemical analysis and organic aerosol source apportionment at nine sites in central Europe: source identification and uncertainty assessment, Atmos. Chem. Phy., 17, 13265-13282, 10.5194/acp-17-13265-2017, 2017.

Emmerson, K. M., Galbally, I. E., Guenther, A. B., Paton-Walsh, C., Guerette, E. A., Cope, M. E., Keywood, M. D., Lawson, S. J., Molloy, S. B., Dunne, E., Thatcher, M., Karl, T., and Maleknia, S. D.: Current

estimates of biogenic emissions from eucalypts uncertain for southeast Australia, Atmos. Chem. Phy., 16, 6997-7011, 10.5194/acp-16-6997-2016, 2016.

Guenther, A. B., Jiang, X., Heald, C. L., Sakulyanontvittaya, T., Duhl, T., Emmons, L. K., and Wang, X.: The Model of Emissions of Gases and Aerosols from Nature version 2.1 (MEGAN2.1): an extended and updated framework for modeling biogenic emissions, Geosci. Model Dev., 5, 1471-1492, 10.5194/gmd-5-1471-2012, 2012.

Hakola, H., Hellén, H., Hemmilä, M., Rinne, J., and Kulmala, M.: In situ measurements of volatile organic compounds in a boreal forest, Atmos. Chem. Phys., 12, 11665-11678, 10.5194/acp-12-11665-2012, 2012.

Hellen, H., Praplan, A. P., Tykka, T., Ylivinkka, I., Vakkari, V., Back, J., Petaja, T., Kulmala, M., and Hakola, H.: Long-term measurements of volatile organic compounds highlight the importance of sesquiterpenes for the atmospheric chemistry of a boreal forest, Atmos. Chem. Phy., 18, 13839-13863, 10.5194/acp-18-13839-2018, 2018.

Heald, C. L., Wilkinson, M. J., Monson, R. K., Alo, C. A., Wang, G. L., and Guenther, A.: Response of isoprene emission to ambient $CO_2$ changes and implications for global budgets, Global Change Biology, 15, 1127-1140, 10.1111/j.1365-2486.2008.01802.x, 2009.

Oderbolz, D. C., Aksoyoglu, S., Keller, J., Barmpadimos, I., Steinbrecher, R., Skjøth, C. A., Plaß-Dülmer, C., and Prévôt, A. S. H.: A comprehensive emission inventory of biogenic volatile organic compounds in Europe: improved seasonality and land-cover, Atmos. Chem. Phys., 13, 1689-1712, 10.5194/acp-13-1689-2013, 2013.

Oikonomakis, E., Aksoyoglu, S., Ciarelli, G., Baltensperger, U., and Prévôt, A. S. H.: Low modeled ozone production suggests underestimation of precursor emissions (especially $NO_x$) in Europe, Atmos. Chem. Phys., 18, 2175-2198, 10.5194/acp-18-2175-2018, 2018.

Poupkou, A., Giannaros, T., Markakis, K., Kioutsioukis, I., Curci, G., Melas, D., and Zerefos, C.: A model for European Biogenic Volatile Organic Compound emissions: Software development and first validation, Environ. Modell. Softw., 25, 1845-1856, 10.1016/j.envsoft.2010.05.004, 2010.

Rinne, J., Ruuskanen, T. M., Reissell, A., Taipale, R., Hakola, H., and Kulmala, M.: On-line PTR-MS measurements of atmospheric concentrations of volatile organic compounds in a European boreal forest ecosystem, Boreal Environment Research, 10, 425-436, 2005.

Sartelet, K. N., Couvidat, F., Seigneur, C., and Roustan, Y.: Impact of biogenic emissions on air quality over Europe and North America, Atmos. Environ., 53, 131-141, 10.1016/j.atmosenv.2011.10.046, 2012.

Schürmann, W.: Emission von Monoterpenen aus Nadeln von Picea Abies (L.) Karst, sowie deren Verhalten in der Atmosphäre, PhD, Technische Universität München, 1993.

Silibello, C., Baraldi, R., Rapparini, F., Facini, O., Neri, L., Brilli, F., Fares, S., Finardi, S., Magliulo, E., Ciccioli, P., and Ciccioli, P.: Modelling of biogenic volatile organic compounds emissions over italy, 18th International Conference on Harmonisation within Atmospheric Dispersion Modelling for Regulatory Purposes (HARMO), Bologna, Italy, 2017.

Viana, M., Hammingh, P., Colette, A., Querol, X., Degraeuwe, B., de Vlieger, I., and van Aardenne, J.: Impact of maritime transport emissions on coastal air quality in Europe, Atmos. Environ., 90, 96-105, 10.1016/j.atmosenv.2014.03.046, 2014.

Tai, A. P. K., Mickley, L. J., Heald, C. L., and Wu, S. L.: Effect of $CO_2$ inhibition on biogenic isoprene emission: Implications for air quality under 2000 to 2050 changes in climate, vegetation, and land use, Geophys. Res. Lett., 40, 3479-3483, 10.1002/grl.50650, 2013.

---

## Author Comment (AC2) · 28 Dec 2018

**Responses to the comments of anonymous referee #2**

We thank the referee for the valuable comments which have greatly helped us to improve the manuscript. Please find below our responses (in black) after the referee comments (in blue). The changes in the revised manuscript are written in *italic*.

GENERAL
Understanding sources of uncertainties in ozone and SOA simulations are important steps for improving air pollution modelings. This study compares two different BVOC schemes and the consequent impacts on ozone and SOA in Europe. The differences between PSI and MEGAN schemes are discussed. The authors found that PSI scheme predicts more monoterpenes while the MEGAN scheme predicts more isoprene. As a result, the CTM based on PSI yields more SOA than the results based on MEGAN scheme. The topic of this study well fits the scope of ACP journal, however, some essential limits may largely weaken the scientific merits of the work.

First, the differences in BVOC are most likely attributed to those in land cover instead of schemes. In general, PSI uses an earlier version of MEGAN parameterization for isoprene and a current MEGAN parameterization for monoterpene. They should have similar responses to environmental factors such as light and temperature. The main reason why PSI and MEGAN schemes show such a large difference in BVOC emissions is that they use different land cover. The authors clarified that PSI is based on tree species while MEGAN is based on PFTs. What if the MEGAN scheme uses the same land cover as PSI, but with PFTs aggregated from tree species? The land cover should be uniform before the comparison.
We agree that different land cover data is one of the major factors leading to differences between the two model outputs. However, the key point is not the land cover data itself, but the corresponding emission factors (PFT-specific for MEGAN, and species-specific for the PSI model). The GlobCover inventory (for the PSI model) is based on the MERIS satellite data obtained by the European Space Agency, while the CLM4-PFTs are derived from MODIS satellite data. In spite of different processing methods for the final data products, we do not expect a large difference in the coverage of broadleaved and needle-leaved forests.

Second, no BVOC observations are used to constrain simulations. Though the authors use the measurements of ozone and OA to validate model results, these are not the direct observations of BVOC. One can simulate right air pollution with wrong reasons (e.g., poor model performance, incorrect meteorology and so on). The only way to check the validity of BVOC schemes is to compare simulations with direct measurements of isoprene and/or monoterpene, which I believe there are many over Europe. Without BVOC constraints, the current study is more like a sensitivity test of ozone and SOA in CTM to any perturbations in BVOC emissions.
For the evaluation of the model we prefer to do this on more stable species like the secondary products such as ozone and SOA. The BVOC concentrations are strongly influenced by local mixing processes and chemistry due to the high reactivity of these molecules. The model output is unlikely representative for species with strong spatial gradients. Also, such measurements are very sparse. In spite of these considerations we agree with the referee that after all it is important to give some idea about the BVOC model performance. We added a figure about the comparison of modelled and measured isoprene in the revised manuscript (see new Fig. 4). The only measurements of some monoterpene species during the simulated period were in Finland. We compared them with our modelled total monoterpene concentrations. We also compared our results with some measurements reported in the literature for other years. We inserted the following text in Section 3.1 (P10, L3-L24) of the revised manuscript:

*„ BVOC measurements are rare and the concentrations are associated with very high spatial gradients (especially vertical) due to high reactivity and local mixing processes that are unlikely captured by the model in the respective grid cell. Nevertheless but with these caveats in mind we compared a few measurements available for isoprene with our model results to get an idea about the range of differences.*

*Compared to monoterpenes, there were more isoprene measurements at various European sites in 2011 (see Fig. 4). Clearly, the MEGAN-isoprene data are much higher than the measurements at all 12 sites while the PSI- isoprene results are closer to the measurements.*

*Unlike the single compound of isoprene, monoterpenes consist of several species and therefore it is even more difficult to perform comparisons with measurements, which are rare and have large uncertainties. Only a limited number of MT measurements were reported in Europe (only in Finland) during our simulation period (Hakola et al., 2012; Hellen et al., 2012). Hakola et al. (2012) reported average MT concentrations of about 508 ppt (with a range between about 150 and 800 ppt) in August 2011 at the SMEAR II station at Hyytiälä. MEGAN-MT for the same period was 117 ppt while PSI-MT was around 2 ppb (for the same site, Rinne et al. (2005) reported MT concentrations of between 200-500 ppt during daytime and more than 1 ppb at nighttime in summer 2004). On the other hand, the measured MT concentrations at a nearby urban background station SMEARIII in Helsinki were lower, with around 117 ppt in summer (Hellen et al., 2012). Both models predicted higher concentrations for that site (MEGAN-MT 303 ppt, PSI-MT 1 ppb). In order to get an idea about the model performance in other regions, we compared our results also with MT concentrations measured at Hohenpeissenberg (southern Germany) in June 2006 (Oderbolz et al., 2013). Both model results (PSI-MT: 75 ppt, MEGAN-MT: 130 ppt) in that region were similar to measurements (~100 ppt). Although this comparison of measurements and model results for different years under different meteorological conditions has a very high uncertainty, it might help to understand the range of differences between the model results and the measurements. In general, all these comparisons suggest that MT concentrations might be underestimated using MEGAN emissions while PSI emissions might be too high over Scandinavia. On the other hand, both models seem to predict MT emissions relatively well in central Europe."*

SPECIFIC

Page 1, line 32: "improving substantially the model performance", How do you know it improves the model for correct reason?

In this manuscript, we only show that the higher monoterpene emissions estimated by the PSI model lead to higher SOA formation and the agreement between the modelled and measured OA improves. In order to understand whether the improvement is due to the biogenic emissions, source apportionment studies are needed. The modelled fraction of biogenic and anthropogenic OA were found to be closer to the PMF analysis of the measured data at Zurich (Canonaco et al., 2013; Daellenbach et al., 2017) when the PSI emissions were used. This is the topic of another manuscript in preparation.

Page 2, Line 38: "highest over all the model inputs" What kind of inputs? Specify.

We rephrased the sentence (Page 3, Line 7-9) as follows:

*"Comparison between MEGAN and another widely used biogenic emission model, the Biogenic Emission Inventory System (BEIS) indicated that the influence of biogenic emission models on ozone simulation results over the United States is far greater than using a different photosynthetically active radiation (PAR) input (Zhang et al., 2017)."*

Page 4, Line 3: "Initial and boundary conditions were: : :" What kind of IC and BC? Specify.

We modified the sentence (Page 4, Lines 23-25) as follows:

*"The gridded initial concentrations of chemical species in each layer of the model domain, as well as at the domain lateral boundaries were obtained from the global model data MOZART-4/GEOS-5 (Horowitz et al., 2003) with a time resolution of 6 hours...."*

Page 5, Line 31: "The value of 0.1 is used for MT in MEGAN2.1, while the values are between 0.065 to 0.077 : : : in PSI model" Why the PSI model uses different parameters while it uses the same scheme as MEGAN?

Although similar response functions are used by PSI model and MEGAN, the PSI model uses species-specific parameters for different monoterpene species historically based on the experimental results reported by Tingey (1980) (e.g. 0.065 is for β-phellandrene, 0.077 is for β-pinene).

Page 6, Line 3: "canopy model" What's the impacts of different canopy models on the simulated light availability for PSI and MEGAN models?

Leaf temperature and PAR in a forest vary substantially within the canopy. Canopy models take into account this effect to calculate the leaf temperature and light for sunlit and shaded layers of the canopy. Since the canopy models of the PSI model and MEGAN are based on similar principles, the effect of different canopy models on the light is not expected to be significant. A BVOC reduction of about 20% due to the canopy model was reported for the PSI model by Oderbolz et al. (2013), however, the influence of different canopy models on BVOC were within the uncertainty range of observed fluxes (Lamb et al., 1996; Guenther et al., 2006). We added a sentence in Page 7, Line 3-5.

*"A BVOC reduction of about 20% due to the canopy model was reported for the PSI model by Oderbolz et al. (2013). Although different canopy models could influence the modelled BVOC emission, such influence was within the uncertainty range of observed fluxes (Lamb et al., 1996; Guenther et al., 2006)."*

Page 6, Line 30: "The variation of biomass density in MEGAN was simulated by the satellite data" How the satellite data simulates biomass density?

We are sorry about the incorrect wording in that sentence. The satellite data of leaf area index is used to quantify the amount and age of foliage for each grid cell via intermediate calculations of the canopy environment model and leaf age model within MEGAN v2.1. We revised the sentence (Page 8, Line 1-2) as follows:

*"The biomass density in MEGAN was calculated by the canopy environment module based on the satellite data of the leaf area index (LAI, $m^2$ leaves per $m^2$ projected area) with a time step of 8 days."*

Page 7, Line 27: "To demonstrate the seasonal differences". The word "demonstrate" is not appropriate, better to use "evaluate" or "quantify".
Done

Page 8, Line 7: "observed" This is not observation. Better to use "found"
Done

Page 8, Line 14: "In winter, highest isoprene emissions occurred in Central Europe for PSI model" Why there are isoprene emissions in winter when leaf biomass is set to zero.
We set the leaf biomass of only deciduous trees to zero but there are some coniferous species emitting isoprene, although the emissions are very low.

Page 9, Line 13: "observed", again not observation. Better to use "calculated"
Corrected

Page 9, Line 28: "background", where is the background ozone from?
The background ozone is the fraction of ozone that is not attributed to local anthropogenic sources and might originate from both natural and anthropogenic sources, such as stratospheric intrusion, long-range transport of ozone from distant areas or production from methane emitted from swamps and wetlands (Vingarzan, 2004). In our simulations, background ozone in the model domain was provided as initial (in

the domain) and boundary concentrations (at the lateral domain boundaries) by the global model MOZART. We revised the sentence in Page 12, Line 2-3.

*"An additional reason might be the rather low ozone production compared to the background ozone where the latter is not affected by local European emissions."*

Page 10, Line 11: "and" should be "but"
Corrected

Page 11, Section 3.2.3: Not sure whether this section is necessary as BVOC has minor impacts on SIA
Although the overall influence of BVOC emissions on SIA is much smaller than on OA, it could reach up to 15% for particulate nitrate on the local scale. Their effect could be even higher in hourly time resolution (analyses were based on monthly average in this study) and under different meteorological conditions (Aksoyoglu et al., 2017). Therefore, we think the results are still important to understand the possible factors influencing the model performance on SIA simulation. Also, we added a sentence about the necessity to investigate the effects of BVOC emissions on SIA in the Introduction (Page 3, Line 17-20).

*"Moreover, BVOCs also influence the secondary inorganic aerosol formation by changing the oxidant concentrations (Aksoyoglu et al., 2017; Karambelas, 2013; Sotiropoulou et al., 2004; Zhang et al., 2016). Aksoyoglu et al. (2017) found that doubled BVOC emissions in Europe led to an increase of particulate inorganic nitrate concentrations by up to 35%."*

Figures 3 and 4: Why in CE, isoprene is much higher for PSI but ozone is still lower than MEGAN?
We think that there might be some misunderstanding. Isoprene by the PSI model in CE is lower than MEGAN in Figure 3. Please note that the scale of the left axis (for PSI emissions) is different from that of the right axis (for MEGAN emissions). To avoid misunderstandings, we updated the figure to have the same scales for the left and right axes for isoprene in summer, where it is seen that MEGAN-isoprene is higher than PSI-isoprene.

[Figure]

**References**

Aksoyoglu, S., Ciarelli, G., El-Haddad, I., Baltensperger, U., and Prévôt, A. S. H.: Secondary inorganic aerosols in Europe: sources and the significant influence of biogenic VOC emissions, especially on ammonium nitrate, Atmos. Chem. Phy., 17, 7757-7773, 10.5194/acp-17-7757-2017, 2017.

Canonaco, F., Crippa, M., Slowik, J. G., Baltensperger, U., and Prévôt, A. S. H.: SoFi, an IGOR-based interface for the efficient use of the generalized multilinear engine (ME-2) for the source apportionment: ME-2 application to aerosol mass spectrometer data, Atmos. Meas. Tech., 6, 3649-3661, 10.5194/amt-6-3649-2013, 2013.

Daellenbach, K. R., Stefenelli, G., Bozzetti, C., Vlachou, A., Fermo, P., Gonzalez, R., Piazzalunga, A., Colombi, C., Canonaco, F., Hueglin, C., Kasper-Giebl, A., Jaffrezo, J. L., Bianchi, F., Slowik, J. G., Baltensperger, U., El-Haddad, I., and Prévôt, A. S. H.: Long-term chemical analysis and organic aerosol source apportionment at nine sites in central Europe: source identification and uncertainty assessment, Atmos. Chem. Phy., 17, 13265-13282, 10.5194/acp-17-13265-2017, 2017.

Guenther, A., Karl, T., Harley, P., Wiedinmyer, C., Palmer, P. I., and Geron, C.: Estimates of global terrestrial isoprene emissions using MEGAN (Model of Emissions of Gases and Aerosols from Nature), Atmos. Chem. Phy., 6, 3181-3210, 10.5194/acp-6-3181-2006, 2006.

Hakola, H., Hellén, H., Hemmilä, M., Rinne, J., and Kulmala, M.: In situ measurements of volatile organic compounds in a boreal forest, Atmos. Chem. Phys., 12, 11665-11678, 10.5194/acp-12-11665-2012, 2012.

Hellen, H., Tykka, T., and Hakola, H.: Importance of monoterpenes and isoprene in urban air in northern Europe, Atmos. Environ., 59, 59-66, 10.1016/j.atmosenv.2012.04.049, 2012.

Horowitz, L. W., Walters, S., Mauzerall, D. L., Emmons, L. K., Rasch, P. J., Granier, C., Tie, X. X., Lamarque, J. F., Schultz, M. G., Tyndall, G. S., Orlando, J. J., and Brasseur, G. P.: A global simulation of tropospheric ozone and related tracers: Description and evaluation of MOZART, version 2, J. Geophys. Res.-Atmos, 108, 10.1029/2002jd002853, 2003.

Karambelas, A.: The interactions of biogenic and anthropogenic gaseous emissions with respect to aerosol formation in the united states, Master of Science, Department of Atmospheric and Oceanic Sciences, University of Wisconsin, Madison, 2013.

Lamb, B., Pierce, T., Baldocchi, D., Allwine, E., Dilts, S., Westberg, H., Geron, C., Guenther, A., Klinger, L., Harley, P., and Zimmerman, P.: Evaluation of forest canopy models for estimating isoprene emissions, J. Geophys. Res.-Atmos, 101, 22787-22797, 10.1029/96jd00056, 1996.

Oderbolz, D. C., Aksoyoglu, S., Keller, J., Barmpadimos, I., Steinbrecher, R., Skjøth, C. A., Plaß-Dülmer, C., and Prévôt, A. S. H.: A comprehensive emission inventory of biogenic volatile organic compounds in Europe: improved seasonality and land-cover, Atmos. Chem. Phys., 13, 1689-1712, 10.5194/acp-13-1689-2013, 2013.

Rinne, J., Ruuskanen, T. M., Reissell, A., Taipale, R., Hakola, H., and Kulmala, M.: On-line PTR-MS measurements of atmospheric concentrations of volatile organic compounds in a European boreal forest ecosystem, Boreal Environment Research, 10, 425-436, 2005.

Sotiropoulou, R. E. P., Tagaris, E., Pilinis, C., Andronopoulos, S., Sfetsos, A., and Bartzis, J. G.: The BOND project: Biogenic aerosols and air quality in Athens and Marseille greater areas, J. Geophys. Res.-Atmos, 109, 16, 10.1029/2003jd003955, 2004.

Tingey, D. T., Manning, M., Grothaus, L. C., and Burns, W. F.: Influence of light and temperature on monoterpene emission rates from slash pine, Plant Physiology, 65, 797-801, 10.1104/pp.65.5.797, 1980.

Vingarzan, R.: A review of surface ozone background levels and trends, Atmos. Environ., 38, 3431-3442, 10.1016/j.atmosenv.2004.03.030, 2004.

Zhang, R., Cohan, A., Biazar, A. P., and Cohan, D. S.: Source apportionment of biogenic contributions to ozone formation over the United States, Atmos. Environ., 164, 8-19, 10.1016/j.atmosenv.2017.05.044, 2017.

Zhang, Y., He, J., Zhu, S., and Gantt, B.: Sensitivity of simulated chemical concentrations and aerosol-meteorology interactions to aerosol treatments and biogenic organic emissions in WRF/Chem, J. Geophys. Res.-Atmos, 121, 6014-6048, 10.1002/2016jd024882, 2016.

---

## Author Comment (AC3) · 28 Dec 2018

**Responses to the comments of anonymous referee #3**

We thank the referee for the valuable comments which helped us to improve the manuscript significantly. Please find below our responses (in black) after the referee comments (in blue). The changes in the revised manuscript are written in *italic*.

Review Summary
Jiang et al. simulated ozone and aerosol concentrations in Europe using two different biogenic emission models (PSI and MEGAN) to probe uncertainties in regional air quality models. They compared model results with ozone observations from the European air quality database, AirBase, and aerosol observations from eight different measurement locations with an Aerodyne AMS or ACSM. Results were generally consistent with previously published papers demonstrating that MEGAN tends to over-estimate isoprene and under-estimate monoterpene emissions. They also found that the simulated ozone mixing ratios between the model runs varied less than the isoprene emissions. This is also consistent with previous studies showing much of Europe's ozone production is NOx-limited rather than VOC-limited. Finally, their model comparison suggests higher monoterpene emissions lead to better comparison between simulated and observed organic aerosol. The authors acknowledge this could be due to compensating factors (e.g. they could be "right for the wrong reasons"). Overall, the scientific approach is reasonable and the scientific questions are appropriate for the scope of the journal. However, it is unclear what information this paper is adding to the scientific community that has not already been published in previous papers. There are also a number of gaps in the methods section that lack clarity. I recommend publication after the manuscript is revised to address the following comments.

General Comments
The authors should better clarify how this particular paper is filling in gaps that have not already been addressed in previous publications. All results sections generally state the results are consistent with work that has already been published, and so it is very unclear what the conclusions from this paper are adding to the growing body of scientific knowledge. The manuscript could better highlight how this work is filling in unique gaps in understanding.

Although our results are consistent with previous studies in general, we think that they provide much more additional information. To our knowledge, there are only a few studies comparing emissions from different BVOC models (Karl et al., 2009; Keenan et al., 2009; Steinbrecher et al., 2009), but comprehensive studies showing the impacts of using different BVOC emission models on secondary pollutants in Europe are scarce. Some studies report the effect of biogenic emissions with zero-out simulations (Sartelet et al., 2012) or with doubled BVOC emissions (Aksoyoglu et al., 2017; Ciarelli et al., 2016). Curci et al. (2009) compared effects of two different biogenic emission inventories, one based on Guenther et al. (1995) and one based on Steinbrecher et al. (2009), on ozone in Europe during 1997 to 2003. However, the limitation of ozone production might have been altered due to large emission reductions of the various precursors in Europe during the past decades.

Our main goal in this study is not just to compare two BVOC models but rather to show how using different BVOC emissions affect the modelled secondary pollutant concentrations and how the effects change spatially and temporally. We chose MEGAN since it is the most widely used biogenic model globally, and the PSI model (which was developed originally for Switzerland and updated for the European domain) to represent models developed specifically for a regional scale. We investigated the effects of using different BVOC emissions not only during summer periods but throughout the whole year. We believe that the OA evaluation with a wide coverage of existing ACSM/AMS measurements in Europe during the simulation period provides valuable information about the influence of BVOC emissions in different parts of Europe in different seasons. In this way, we also want to emphasize the need to harmonize the biogenic emissions as much as possible in model inter-comparison studies. Although their importance on air quality modeling results are well known, BVOC emissions are usually not prescribed in model inter-comparison studies (e.g. AQMEII, Eurodelta, MICS-Asia) making it very difficult to compare and interpret the results.

Furthermore, although the effects of different BVOC emissions on ozone have been reported in a few previous studies, it is important to keep the knowledge updated in the context of continuous reduction of anthropogenic emissions since 1990s, which could change the sensitivity of secondary pollutants formation to precursor emissions in some regions. We revised the Introduction to make the objective and novelty of this study clearer as follows:

Page 2, Line 9-18:
*Although there are a few studies comparing different BVOC models (Steinbrecher et al., 2009; Karl et al., 2009; Keenan et al., 2009), comprehensive studies showing the impacts of using different BVOC emission models on secondary pollutants in Europe are scarce. Some studies report the effect of biogenic emissions with zero-out simulations (Sartelet et al., 2012) or with doubled BVOC emissions (Aksoyoglu et al., 2017). Curci et al. (2009) compared the effects of two different biogenic emission inventories, one based on Guenther et al. (1995) and one based on Steinbrecher et al. (2009), on ozone in Europe during 1997 to 2003. However, the limitation of ozone production might have been altered due to large emission reductions of the various precursors in Europe during the past decades. Understanding the potential influence of biogenic emissions on European air quality is therefore of great importance especially under the continuously reduced anthropogenic emissions since early 1990s.*

Page 3, Line 22-24:
*In spite of an increasing interest in understanding the influence of biogenic emissions on ozone and aerosols, limitations still remain: most of the studies focus on short periods (mostly in summer), while the potential influence of BVOC on SOA could still be high in winter at local scale, the evaluation of modelled OA is challenged by the scarcity of field measurements, and not much attention has been paid to the effects of BVOC on SIA by different biogenic models.*

Page 3, Line 25-29:
*Biogenic emissions in Europe were estimated by two BVOC emission models with different land cover and emission factors; MEGAN as a widely used model globally and the PSI model to represent models developed for a specific region. The BVOC emissions from the two models were then used as input for the regional air quality model Comprehensive Air Quality Model with extensions (CAMx) to simulate gaseous and particulate pollutant concentrations in 2011.*

Specific Comments
METHODS: SECTION 2.2.1 EMISSION RATES Authors state they estimate reference emission rates of isoprene and monoterpenes based on Lamb et al., 1993 but then go on to say Norway spruce isoprene emissions were estimated to be 10% of alpha-pinene. It is unclear why Norway spruce was handled differently and why it is singled out to be described separately from the other plant species. Please clarify.
The PSI model was originally developed in the early 90s only for Switzerland using a very detailed tree inventory. Norway Spruce covers almost half of the Swiss forests (49%) and it is also an abundant forest type in the other European regions, it was therefore treated explicitly using some explicit data available in Europe at that time (Schürmann, 1993; Steinbrecher, 1989).

The PSI model emission rates are species-specific except for "pasture" and "crop" (Table 1). How much variability would you expect between different types of "pasture vegetation" and "crop vegetation" based on the literature? What proportion of the total area covered in the model is characterized as "pasture" and "crop"? Is it a significant portion of the land that could drastically impact results or is it minor?
Although the coverage of crop and pasture is large (see Fig. 1 below), their contribution to total BVOC emissions is small because of their low biomass density, emission rates and short vegetation period compared to forests (Simpson et al., 1999). Therefore, we believe that the impact of different types of pasture and crops on the results is minor.

[Figure]

**Fig.1:** Fraction of agricultural land (left) and pasture (right) in the model domain.

Sesquiterpene emissions: authors state that sesquiterpene emissions were assumed to be 5% (by weight) of monoterpene emissions based on field measurements from various studies, but then cite a single paper that is actually a modelling paper and not a review or synthesis of measurements. Please cite the original literature from which this "5% (by weight)" reference is derived.

Thank you for this comment. The approximation of sesquiterpene emissions as 5% of monoterpenes was estimated using the data compiled from various emission databases containing both monoterpene and sesquiterpene emission rates for 116 tree species (Steinbrecher et al., 2009, Suppl.). We revised the sentence about sesquiterpenes (Page 5, Line 25-27) as follows:

*"...SQT emissions were treated only as pool emissions and assumed to be 5% (by weight) of the monoterpene emissions based on the emission rate data for 116 species compiled from various studies as given by Steinbrecher et al. (2009)."*

METHODS: SECTION 2.2.2, RESPONSE FUNCTIONS If sesquiterpenes are being treated as pooled emissions as stated in the previous section, then they will be treated similar to monoterpenes. However, the authors do not discuss what value they used for sesquiterpenes. In Guenther et al., (2012) the empirically-derived temperature correction coefficient, for sesquiterpenes was 0.17. Was that the value used in this study as well? Please clarify.

As we stated before, sesquiterpene emissions in PSI model were not calculated explicitly, but their emission rates were scaled to the monoterpene pool emissions. Therefore, they were treated similarly to the monoterpene emissions (5% (by weight) of the monoterpene emissions, as stated above).

Again, the authors single out Norway spruce emissions being handled a bit differently than other plant species. In this case, the Norway spruce monoterpene emissions have some light-dependent fraction estimated based on a study in 1993. Why is Norway spruce being singled out for more detailed emission estimation? Is it the dominant species in the modeling domain? This should be clarified.

The Norway spruce (*picea abies*) is indeed the most typical tree species in northern and central Europe. As we explained above, the PSI model was originally developed in the early 90s for Switzerland. Norway Spruce covers almost half of the Swiss forests (49%) and it is also an abundant forest type in the other European regions, it was therefore treated explicitly using data from Norway spruce studies.

Also, more details should be included about how the light-dependent emissions were estimated instead of just referring to the 1993 paper with no summary of what information was taken from that paper and used in this study. Finally, does this section then imply that all other monoterpene emissions were light-independent? Can this be stated more clearly and justified? If all monoterpene emissions are being treated as light-independent (except for some unstated fraction of Norway spruce monoterpene emissions), then this should be justified because it is well known that a substantial fraction of monoterpene emissions are

lightdependent; for example, in MEGANv2.1 the light-dependent fraction of monoterpene emissions ranges from 40-80%! (see Guenther et al., 2012, Table 4).

We apologize for the ambiguity in this issue. We referred to Guenther et al. (2012) in Page 6, lines 1-2 for MEGAN. In the PSI model, light-dependent MT emissions were calculated as a function of PAR for all the individual monoterpenes emitted from Norway spruce. We updated this section in the revised manuscript (Page 6, Line 19-22) as follows:

*"The light-dependent synthesis emissions of MTs were considered in MEGAN v2.1 as described in Guenther et al. (2012). Depending on different MT species, the light-dependent fraction of MT emissions ranges between 0.2 to 0.8 for MEGAN. In the PSI model the light-dependent emissions from Norway spruce are calculated for each monoterpene species as a function of PAR based on the data of Schürmann (1993)."*

METHODS SECTION 2.2.3 INPUTS OF DRIVING VARIABLES Unclear how GlobCover 2006 data is being used to derive species-level distributions. How did the authors go from fractions of "needleleaf, broadleaf and mixed forests" to plant species distribution using the profiles from Simpson et al., 1999? There is missing information here that links the two.

We added the detailed procedures describing the calculation of the species-level distribution based on GlobCover 2006 data and Simpson's profile in Page 7, Line 27-32.

*"The original 35 forest species in Simpson et al. (1999) were grouped into 10 classes (including 5 coniferous and 5 broadleaf species), and the ratio of each species class to the total coniferous forest and broadleaf forest was calculated (Table S2). The ratio of "other trees" were proportionally added to the 10 tree species. As the "other trees" are mainly found in a few Mediterranean countries, their influence on the whole domain is small. The species-coverage was then generated by multiplying the forest coverage from GlobCover with the country-specific tree species profile."*

How much of variation between PSI and MEGAN emissions was driven by differences in normalizing emissions to leaf surface area (MEGAN) versus leaf biomass (PSI)? Are there potential biases that could vary between plant types for comparing total canopy-scale flux that arise from how surface area versus biomass are scaled up?

The PSI model estimates the plant-specific emissions and it uses the biomass densities (g m$^{-2}$) to convert the emission factors of specific plant species (in µg g$^{-1}$ h$^{-1}$) based on Steinbrecher et al. (2009) to emission rates in µg m$^{-2}$ h$^{-1}$. On the other hand, MEGAN emission factors for each PFT are given directly in µg m$^{-2}$ h$^{-1}$. At the end, the emission rates in both models are in µg m$^{-2}$ h$^{-1}$. A direct comparison is not possible because of different modeling approaches. Differences might arise also from using different land use data, different emission factors and different biomass densities. However, a detailed comparison of BVOC models is out of scope of this manuscript. We focus here on the effects of using different BVOC models on modelled secondary pollutant concentrations.

How did the authors ensure they were making meaningful comparisons between the models with the emissions normalized differently? Figure 2 was clear: authors graphed the emission rate per model grid cell. It was less clear how this comparison was done in Figure 3 where the graph simply shows the emission rate. Was this also per model grid? Per entire modeling domain? This should be stated more clearly.

The Figure 3 was based on the average emissions per grid cell in the entire model domain. We revised the units of the y-axis labels (kg cell$^{-1}$ h$^{-1}$), and updating the caption to *"Diurnal variations of average grid-scale isoprene and monoterpene emissions in the model domain"*.

Figure 2: right axis label is cut off on third row.
Corrected.

The Figure 5 (in the revised manuscript) caption was updated as:

*"Mean bias of surface $O_3$ mixing ratios in the afternoon (12:00–18:00 UTC) for each bin of observed hourly average ozone in July 2011."*

RESULTS $O_3$: results are consistent with previous literature demonstrating that $O_3$ production in most of Europe is in a $NO_x$-limited regime as opposed to a VOC-limited regime and thus the isoprene differences between the two models do not translate into large differences in ozone. Not a novel result. Can the authors comment on how this study is different from previous ones that have published the same result?

Although in the past ozone formation was more sensitive to $NO_x$ in most of Europe, we think that it might be changing and would be different on a local scale as a result of large emission reductions since the 1990s. Our results suggest that the regions that are affected more by higher isoprene emissions from MEGAN are especially around the coastal regions in the south (see Fig. 5, right panel) where isoprene emissions are relatively higher than in other regions, but also where $NO_x$ emissions from shipping are still high (not regulated as land emissions by the revised Gothenburg Protocol). It is therefore not clear how ozone formation will evolve with reduced land emissions while ship emissions continuously increase especially around the Mediterranean. We deepened the discussion by adding the regional analysis based on previous studies in P11, L33 to P12, L11.

*"The main reason for the weak effect of the isoprene emissions on ozone is the stronger sensitivity of ozone formation in general to $NO_x$ emissions rather than VOC emissions in Europe. An additional reason might be the rather low ozone production compared to the background ozone where the latter is not affected by local European emissions (Oikonomakis et al., 2018; Sartelet et al., 2012). Several European studies reported that ozone formation in most regions is $NO_x$-sensitive except around the English Channel, Benelux and Po Valley regions, where $NO_x$ emissions are high (due to intensive anthropogenic NOx emissions from both land and shipping or geographical characteristics leading to high accumulation of pollutants) and the response to a change in the VOC emissions is relatively stronger (Aksoyoglu et al., 2012; Beekmann and Vautard, 2010; Oikonomakis et al., 2018). However, the sensitivity of ozone formation to its precursor emissions might change as a result of large $NO_x$ emission reductions in Europe since 1990 according to the Gothenburg Protocol. On the other hand, emissions from shipping activities are not regulated as strictly as land emissions and have been increasing continuously especially in the Mediterranean, affecting both ozone and particulate matter concentrations (Aksoyoglu et al., 2016; Viana et al., 2014)."*

OA comparison: the study does not have a single AMS/ACSM location in the Northern Europe region. Surely there are measurements at Hyytiälä, or some other boreal forest site in northern Europe. Can you justify why no measurement sites were included for northern Europe?

We agree that comparison with measurements in Northern Europe (where the difference in emissions between the two BVOC models is largest) is important. However, OA measurements in Northern Europe are quite scarce. Although the AMS/ACSM stations of Hyytiälä and Vavihill did not have data available for the period of interest, there was one dataset available from a campaign at SMEAR II station at Hyytiälä between 15 March and 20 April 2011 (Kortelainen et al., 2017). The comparison of modelled OA by both PSI and MEGAN emissions with that dataset showed that the modelled OA could capture the temporal variation of measurements and PSI emissions led to a better agreement between modelled and measured OA (see Fig. 2 below). The comparison of daily OA at SMEARII were added to Figure 7, and the statistical results of the new stations were added to Table 3.

[Figure]

**Fig. 2:** Comparison of measured and modelled OA at SMEARII Hyytiälä station.

SECTION 3.2.3: INORGANIC AEROSOLS Authors do not set up a rationale in the introduction for investigating the impact of changing biogenic emissions on inorganic aerosols. Why would differences in biogenic emissions substantially alter inorganic aerosol? Without this rationale in the introduction, this section does not fit with the rest of the paper.

Isoprene and monoterpenes react with oxidants such as OH, ozone and $NO_3$ in the atmosphere and therefore they might lead to changes in oxidant concentrations, which are also involved in the formation of secondary inorganic aerosols such as ammonium nitrate and sulfate. Although such effects are smaller than the effects on organic aerosols, we think that it is worth including them. We have updated the introduction (in Page 3, Line 17-20) to highlight the rationale to study the impact of BVOCs input on inorganic aerosols.

*"Moreover, the BVOCs also influence the secondary inorganic aerosol formation by changing the oxidant concentrations (Aksoyoglu et al., 2017; Karambelas, 2013; Sotiropoulou et al., 2004; Zhang et al., 2016). Aksoyoglu et al. (2017) found that doubled BVOC emissions in Europe led to an increase of particulate inorganic nitrate concentrations by up to 35%."*

DISCUSSION Authors end the paper by saying, "In future studies, BVOC emission models with more regional specific adaptation in vegetation types and emission factors are urgently needed to reduce the uncertainties in BVOC emission estimates in order to improve air quality modelling." Why is this the recommendation rather than simply improving the emission factors for the plant functional types in MEGAN? It is not reasonable to model the emissions from every single plant species on the planet. I don't agree that the results from this study emphasize the need for more plant specificity because even this paper only used a sub-set of 10 specific plant species (with an additional two general classes for "pasture" and "crop"). It seems to me that the major finding from this paper, consistent with published papers before it, is that the MEGAN emission factors could be updated and improved.

We agree with the referee that it is not reasonable to model the emissions from each single plant species. Our point is that the emission factors need to be improved based on the regional information such as vegetation types (for MEGAN). However, this suggestion is not specific for MEGAN, but also for similar species-specific models like the PSI model. As the referee noted, only 10 specific trees were included in the PSI model (they were originally selected according to the forest composition in Switzerland, they are however typical also for Europe), which should be improved in the future. We have revised that paragraph in Page 15, L24-26.

*"In future studies, emission factors should be improved in BVOC models to include more regional specific vegetation types to reduce the uncertainties in BVOC emission estimates and to improve air quality modeling results."*

**Reference**

Aksoyoglu, S., Keller, J., Oderbolz, D. C., Barmpadimos, I., Prévôt, A. S. H., and Baltensperger, U.: Sensitivity of ozone and aerosols to precursor emissions in Europe, Int. J. Environ. Pollut., 50, 451-459, 10.1504/ijep.2012.051215, 2012.

Aksoyoglu, S., Baltensperger, U., and Prevot, A. S. H.: Contribution of ship emissions to the concentration and deposition of air pollutants in Europe, Atmos. Chem. Phy., 16, 1895-1906, 10.5194/acp-16-1895-2016, 2016.

Aksoyoglu, S., Ciarelli, G., El-Haddad, I., Baltensperger, U., and Prévôt, A. S. H.: Secondary inorganic aerosols in Europe: sources and the significant influence of biogenic VOC emissions, especially on ammonium nitrate, Atmos. Chem. Phy., 17, 7757-7773, 10.5194/acp-17-7757-2017, 2017.

Beekmann, M., and Vautard, R.: A modelling study of photochemical regimes over Europe: robustness and variability, Atmos. Chem. Phy., 10, 10067-10084, 10.5194/acp-10-10067-2010, 2010.

Ciarelli, G., Aksoyoglu, S., Crippa, M., Jimenez, J. L., Nemitz, E., Sellegri, K., Äijälä, M., Carbone, S., Mohr, C., O'Dowd, C., Poulain, L., Baltensperger, U., and Prévôt, A. S. H.: Evaluation of European air quality modelled by CAMx including the volatility basis set scheme, Atmos. Chem. Phy., 2016, 10313-10332, 10.5194/acpd-15-35645-2015, 2016.

Curci, G., Beekmann, M., Vautard, R., Smiatek, G., Steinbrecher, R., Theloke, J., and Friedrich, R.: Modelling study of the impact of isoprene and terpene biogenic emissions on European ozone levels, Atmos. Environ., 43, 1444-1455, 10.1016/j.atmosenv.2008.02.070, 2009.

Guenther, A., Hewitt, C. N., Erickson, D., Fall, R., Geron, C., Graedel, T., Harley, P., Klinger, L., Lerdau, M., McKay, W. A., Pierce, T., Scholes, B., Steinbrecher, R., Tallamraju, R., Taylor, J., and Zimmerman, P.: A global-model of natural volatile organic-compound emissions, J. Geophys. Res.-Atmos, 100, 8873-8892, 10.1029/94jd02950, 1995.

Guenther, A. B., Jiang, X., Heald, C. L., Sakulyanontvittaya, T., Duhl, T., Emmons, L. K., and Wang, X.: The Model of Emissions of Gases and Aerosols from Nature version 2.1 (MEGAN2.1): an extended and updated framework for modeling biogenic emissions, Geosci. Model Dev., 5, 1471-1492, 10.5194/gmd-5-1471-2012, 2012.

Hakola, H., Hellén, H., Hemmilä, M., Rinne, J., and Kulmala, M.: In situ measurements of volatile organic compounds in a boreal forest, Atmos. Chem. Phys., 12, 11665-11678, 10.5194/acp-12-11665-2012, 2012.

Hellen, H., Tykka, T., and Hakola, H.: Importance of monoterpenes and isoprene in urban air in northern Europe, Atmos. Environ., 59, 59-66, 10.1016/j.atmosenv.2012.04.049, 2012.

Horowitz, L. W., Walters, S., Mauzerall, D. L., Emmons, L. K., Rasch, P. J., Granier, C., Tie, X. X., Lamarque, J. F., Schultz, M. G., Tyndall, G. S., Orlando, J. J., and Brasseur, G. P.: A global simulation of tropospheric ozone and related tracers: Description and evaluation of MOZART, version 2, J. Geophys. Res.-Atmos, 108, 10.1029/2002jd002853, 2003.

Karambelas, A.: The interactions of biogenic and anthropogenic gaseous emissions with respect to aerosol formation in the united states, Master of Science, Department of Atmospheric and Oceanic Sciences, University of Wisconsin, Madison, 2013.

Karl, M., Guenther, A., Köble, R., Leip, A., and Seufert, G.: A new European plant-specific emission inventory of biogenic volatile organic compounds for use in atmospheric transport models, Biogeosciences, 6, 1059-1087, 10.5194/bg-6-1059-2009, 2009.

Keenan, T., Niinemets, U., Sabate, S., Gracia, C., and Penuelas, J.: Process based inventory of isoprenoid emissions from European forests: model comparisons, current knowledge and uncertainties, Atmos. Chem. Phy., 9, 4053-4076, 10.5194/acp-9-4053-2009, 2009.

Kortelainen, A., Hao, L. Q., Tiitta, P., Jaatinen, A., Miettinen, P., Kulmala, M., Smith, J. N., Laaksonen, A., Worsnop, D. R., and Virtanen, A.: Sources of particulate organic nitrates in the boreal forest in Finland, Boreal Environment Research, 22, 13-26, 2017.

Oderbolz, D. C., Aksoyoglu, S., Keller, J., Barmpadimos, I., Steinbrecher, R., Skjøth, C. A., Plaß-Dülmer, C., and Prévôt, A. S. H.: A comprehensive emission inventory of biogenic volatile organic compounds in Europe: improved seasonality and land-cover, Atmos. Chem. Phys., 13, 1689-1712, 10.5194/acp-13-1689-2013, 2013.

Oikonomakis, E., Aksoyoglu, S., Ciarelli, G., Baltensperger, U., and Prévôt, A. S. H.: Low modeled ozone production suggests underestimation of precursor emissions (especially NOx) in Europe, Atmos. Chem. Phys., 18, 2175-2198, 10.5194/acp-18-2175-2018, 2018.

Sartelet, K. N., Couvidat, F., Seigneur, C., and Roustan, Y.: Impact of biogenic emissions on air quality over Europe and North America, Atmos. Environ., 53, 131-141, 10.1016/j.atmosenv.2011.10.046, 2012.

Schürmann, W.: Emission von Monoterpenen aus Nadeln von Picea Abies (L.) Karst, sowie deren Verhalten in der Atmosphäre, PhD, Technische Universität München, 1993.

Simpson, D., Winiwarter, W., Borjesson, G., Cinderby, S., Ferreiro, A., Guenther, A., Hewitt, C. N., Janson, R., Khalil, M. A. K., Owen, S., Pierce, T. E., Puxbaum, H., Shearer, M., Skiba, U., Steinbrecher, R., Tarrason, L., and Oquist, M. G.: Inventorying emissions from nature in Europe, J. Geophys. Res.-Atmos, 104, 8113-8152, 10.1029/98jd02747, 1999.

Sotiropoulou, R. E. P., Tagaris, E., Pilinis, C., Andronopoulos, S., Sfetsos, A., and Bartzis, J. G.: The BOND project: Biogenic aerosols and air quality in Athens and Marseille greater areas, J. Geophys. Res.-Atmos, 109, 16, 10.1029/2003jd003955, 2004.

Steinbrecher, R.: Gehalt und Emission von Monoterpenen in oberirdischen Organen von Picea Abies, Ph.D, Technische Universitat Miinchen, 1989.

Steinbrecher, R., Smiatek, G., Koble, R., Seufert, G., Theloke, J., Hauff, K., Ciccioli, P., Vautard, R., and Curci, G.: Intra- and inter-annual variability of VOC emissions from natural and semi-natural vegetation in Europe and neighbouring countries, Atmos. Environ., 43, 1380-1391, 10.1016/j.atmosenv.2008.09.072, 2009.

Viana, M., Hammingh, P., Colette, A., Querol, X., Degraeuwe, B., de Vlieger, I., and van Aardenne, J.: Impact of maritime transport emissions on coastal air quality in Europe, Atmos. Environ., 90, 96-105, 10.1016/j.atmosenv.2014.03.046, 2014.

Zhang, Y., He, J., Zhu, S., and Gantt, B.: Sensitivity of simulated chemical concentrations and aerosol-meteorology interactions to aerosol treatments and biogenic organic emissions in WRF/Chem, J. Geophys. Res.-Atmos, 121, 6014-6048, 10.1002/2016jd024882, 2016.